

# Assessment of object-based indices to identify convective organization

Giulio Mandorli[1] and Claudia J. Stubenrauch[1]

[1]Laboratoire de Météorologie Dynamique / Institut Pierre-Simon Laplace, (LMD/IPSL), Sorbonne Université, Ecole Polytechnique, CNRS, Paris, France

**Correspondence:** Giulio Mandorli (giulio.mandorli@lmd.ipsl.fr)

**Abstract.**

The emerging field of convective organization has attracted significant attention due to its potential implications for weather and climate. Numerous indices have been developed to identify organization of convection, serving as essential tools for advancing our understanding in this area. Because of the large number of convective indices, many results on organization are
still uncertain and different studies have shown diverging results. The present analysis studies and compares nine object-based indices in order to evaluate their ability to quantify organization. The analysis begins by establishing a set of criteria expected for convective organization and subsequently subjecting the indices to assessment against these benchmarks. The criteria are organized into three categories. The first category tests the robustness of the indices against noise. The second category evaluates their sensitivity to the size and position of the convective objects. The third category assesses their dependency on
the specific characteristics of the dataset in use. Among the indices scrutinized, none fulfill all the desired conditions, and some conditions are only marginally satisfied. Therefore, we developed a new index, called Organization Index based on Distance and Relative Area (OIDRA), as an example of a well-behaving index. The unmet conditions and differences between indices can explain the discord between different organization studies. The results come down to a guideline that will help to advance our description of deep convective organization.

## 1 Introduction

Convection is the main responsible for transporting heat and moisture through the atmosphere. Thus it drives Earth's weather and climate. In particular, in the tropics, convection may appear either isolated or clustered. In this second case, we talk about aggregated or organized convection, or convective organization. Because of the great importance of convection on climate, lately, many studies have been focusing their attention on convective organization and its implications, using both simulations
and observations (e.g., Wing et al., 2017, 2020; Bony et al., 2020; Bläckberg and Singh, 2022; Stubenrauch et al., 2023). However, quantifying the degree of convective organization is challenging and there is still no consensus on the best method to use. Various methods have been proposed to quantify the degree of organization in recent years (Biagioli and Tompkins, 2023). These organization metrics are often given by a single real number, called organization index. The recently emerging large number of convective indices and diverging results (Wing et al., 2017; Stubenrauch et al., 2023) ask for a systematic





assessment, i.e. statistical studies that verify the robustness of these indices. This is challenging because convective organization has not a rigorous definition. Nevertheless, it is still possible to verify if these indices satisfy certain conditions expected from the metrics of organization. Such studies have been so far performed only for example cases.

Assessing metrics via verifying some conditions has already been done in other fields, for example in high-energy physics by Cacciari et al. (2008). This approach is reproduced here to refine our definition of convective organization. Hence, this
statistical study assesses nine object-based indices by verifying if they satisfy seven conditions. Only object-based indices are studied in this work, while non-object-based indices are not included.

This article is outlined as follows. Sect. 2.1 describes how we reconstruct the convective objects within a selected region from an existing deep convective cloud tracking database. Moreover, it briefly recalls the convective organization indices under study. Sect. 3 describes the procedure of this study, and section 4 shows the results. Finally, Sect. 5 summarizes the results and
discusses a potential outlook.

## 2    Data and Methods

### 2.1    The TOOCAN dataset

The statistical comparison between indices needs a dataset of horizontal binary fields that mimic deep convective clouds for which it is possible to compute the convective organization indices. Since the goal is not to study physical processes but the
behavior of the indices, any dataset can be used. However, in order to well represent the typical size, occurrence, and disposition of deep convection in the tropics, we have chosen a real satellite dataset with a good spatial and temporal resolution.

The Tracking of organized convection algorithm through 3D segmentation (TOOCAN) (Fiolleau and Roca, 2013) provides calibrated infra-red (IR) brightness temperatures ($T_B$) on the entire tropical band, spanning 30N to 30S (Fiolleau et al., 2020) by combining different geostationary satellites. The algorithm relies on an iterative process involving the detection and expan-
sion of convective seeds in the spatiotemporal domain. Convective seeds are initially defined by $T_B < 190$ K, then they are expanded to 10-connected spatiotemporal neighborhoods, and a selection on the time duration and maximum area is applied. The resulting objects are successively employed as seeds and the procedure is repeated. At each iteration, the $T_B$ threshold is increased by 5 K until the threshold of 235 K is reached. The spatial resolution is 0.04° and the temporal frequency is 30 minutes.

For this study, we selected the oceanic tropical Warm Pool region expanding over 0°N-9.6°N and 140.4°E-150°E. The original resolution is downscaled to 0.08° in order to analyze images with a size of 120x120 grid boxes. In the following, we only use the convective system identification, i.e. the deep convective cloud mask of TOOCAN. We do not use the system identifiers which has been obtained by the tracking. Instead, we reconstruct the convective objects using the grid boxes that are identified as convective systems by TOOCAN, and grouping all 8-connected grid boxes. This procedure is implemented with
the python framework developed by van der Walt et al. (2014). Holes in each object are filled to avoid degenerate dispositions. Then, images with less than two objects are rejected. Finally, a total of 76462 images in the period 2012-2016 is considered for this study.

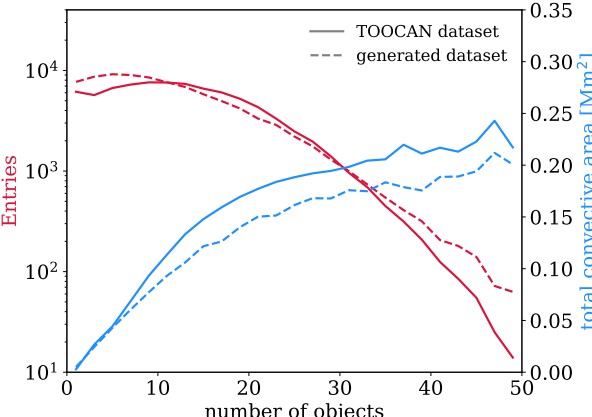

**Figure 1.** Distributions of events and total area of convection as a function of the number of objects for both the TOOCAN dataset and the generated dataset. The horizontal size of the domain is 0.92 Mm$^2$.

Independently, a dataset is generated to compare with the results obtained with the TOOCAN dataset. This dataset is built as images with randomly placed circular objects of different sizes. The sizes and the number of objects are tuned to match with the ones of the TOOCAN dataset, in order to have a peer-to-peer comparison between the two datasets. The results obtained with the two datasets are similar, meaning that they don't depend on the nature of the objects. The results obtained with the TOOCAN database are shown in the following, while the ones obtained with the newly generated dataset are shown in the supplement material. In order to provide a context for this study, the frequency distribution of the number of objects and their total area is shown in Fig. 1. For both datasets, the frequency of the number of objects decreases, while the total area of the convective systems increases with increasing number of objects. Examples of the analyzed images are given in the following sections.

## 2.2 Indices of convective organization

The indices of organization try to measure the degree of organization in a determined domain. These indices are computed by using the size and position of the objects described in the previous section. Using the same data is crucial for performing a comparison because any differences in the organization values come entirely from differences in the indices' behavior.

The index $I_{org}$ was conceptually developed by Weger et al. (1992), for the study of Cumulus cloud fields. They compared the cumulative distribution of the object's nearest neighbor distance (NNCDF) of the cloud disposition with the one of a random disposition. Later, Tompkins and Semie (2017) exploited the idea of comparing the two NNCDFs to extract one single value that discriminates between random and clustered clouds. The $I_{org}$ is a real number between 0 and 1. Values larger than 0.5 are associated with organized convection while values of $I_{org} \lesssim 0.5$ indicate disorganized convection. The comparison between nearest-neighbors makes $I_{org}$ insensitive to organization beyond the $\beta$-mesoscale ($\sim$100 km). Therefore Biagioli and





Tompkins (2023) developed a new index, called $L_{org}$, that is sensitive to all scale organization. $L_{org}$ is computed by comparing the theoretical and observed distributions of all-paired convective object distances. The index $L_{org}$ is dimensionless, it is zero for random disposition and it is positive for organized convection. Both $I_{org}$ and $L_{org}$ do not take into account the size of the objects.

As a consequence, White et al. (2018) developed the Convective Organization Potential (COP) by assuming that 2D objects that are larger and closer together are more likely to interact with each other in the horizontal plane. Its concept reproduces the gravitational potential and it is determined from the distance between the centers of the objects and the radii of equal-area circles A scalar value is associated with each unique connection between pairs of objects, and the arithmetic average is computed over all the pairs. Recently, a modification of COP called Area-based COP (ABCOP), was proposed by Jin et al. (2022) in order to improve the dependency on the object's area and number. Both COP and ABCOP are positive, dimensionless and higher values represent higher degrees of organization.

The Radar Organization Metric (ROME) (Retsch et al., 2020) has been built, similarly to COP and ABCOP, on a sum of scalar values associated with all unique object pairs. Originally, ROME was developed to quantify organization on small domains with high resolution. However, it has also been applied to larger domains (Bläckberg and Singh, 2022; Stubenrauch et al., 2023). The value of ROME is always positive and between one and two times the object's mean area. ROME is measured in km$^2$ (or equivalently, in number of grid boxes) and its value is large for a high degree of organization.

The Simple Convective Aggregation Index (SCAI) was introduced by Tobin et al. (2012) and is based on the number of and distance between convective objects within a domain. This index has been used in studies by Tobin et al. (2012, 2013) and Stein et al. (2017). Since SCAI does not consider the object sizes, the Modified Convective Aggregation Index (MCAI) was proposed by Xu et al. (2019) to correct this feature. These indices are unitless and they are inversely proportional to the total number of grid boxes in the domain. Thus their values scale with the size of the image under consideration, as for ROME. SCAI and MCAI are the only indices that identify organized convection with low values, therefore, in our study, we have negated SCAI and MCAI for an easier comparison with the other indices. Consequently, both negated SCAI and MCAI are always negative and their values are high (close to zero) for organized objects and low for disorganized ones.

The Morphological Index of Convective Aggregation (MICA) (Kadoya and Masunaga, 2018) is the only index that does not consider the relative object's disposition, but it considers the amount of space on the domain where no convection is occurring.

Along with all the above-mentioned organization indices, the degree of organization has also been estimated using just the total area of convection (Tan et al., 2015; Bao et al., 2017), and they are included in this comparison. The number of convective objects within the domain has also been used to quantify convective organization (Tobin et al., 2013; Bläckberg and Singh, 2022). However, in this manuscript, it is not explicitly presented since the associated results closely resemble those obtained with the indices SCAI and MCAI.

The study of the above indices led us to the development of a new index of organization. This index is called Organization Index based on Distance and Relative Area (OIDRA), and it is studied in the next section together with all the other indices.





**Table 1.** Correlation of the indices with each other and with the number of convective objects, the total area of convection, and the mean object size. The correlations are obtained using the object reconstruction from TOOCAN mask.

| | $I_{org}$ | $L_{org}$ | COP | ABCOP | ROME | SCAI | MCAI | MICA | OIDRA | number | total area | mean size |
|---|---|---|---|---|---|---|---|---|---|---|---|---|
| $I_{org}$ | 100 | 74 | 38 | -15 | -25 | 35 | 31 | 43 | 10 | -23 | -33 | -26 |
| $L_{org}$ | 74 | 100 | 47 | -16 | -16 | 41 | 40 | 56 | 22 | -26 | -27 | -16 |
| COP | 38 | 47 | 100 | -1 | 39 | 47 | 50 | 72 | 48 | -43 | 1 | 39 |
| ABCOP | -15 | -16 | -1 | 100 | 47 | -34 | -31 | -13 | 39 | 33 | 81 | 46 |
| ROME | -25 | -16 | 39 | 47 | 100 | 5 | 10 | 14 | 52 | -10 | 68 | 100 |
| SCAI | 35 | 41 | 47 | -34 | 5 | 100 | 99 | 49 | 31 | -96 | -48 | 5 |
| MCAI | 31 | 40 | 50 | -31 | 10 | 99 | 100 | 51 | 34 | -96 | -43 | 10 |
| MICA | 43 | 56 | 72 | -13 | 14 | 49 | 51 | 100 | 49 | -43 | -19 | 13 |
| OIDRA | 10 | 22 | 48 | 39 | 52 | 31 | 34 | 49 | 100 | -29 | 39 | 51 |

The exact definitions of all indices are given in Appendix A1. All the introduced indices have the same goal: identify organized convection and discriminate it from unorganized one. However, as shown by the correlations in Table 1, the indices do not give a coherent answer.

    The correlations between these indices are often smaller than 0.5, and sometimes they are even negative. This means that the estimated strength of organization depends on the index. Hence, the choice of the organization index may strongly affect

the conclusions of climate studies of deep convective organization, in particular when the indices used are anti-correlated.

    Table 1 also presents the correlations between the above indices with three important variables that characterize the convective systems within the studied domain: the mean size of the convective objects ($\sum_i A_i/N$), their number (N), and the total convective area covering the domain ($\sum_i A_i$). Before proceeding with the analysis, we highlight the following correlations:

    – ROME is completely correlated to the mean size of the objects. Thus, ROME reflects the mean object size entirely.

– SCAI and MCAI are strongly correlated to the number of objects.

    – ABCOP strongly correlates to the total area of convection covering the studied domain.

    All the other indices show weaker relationships with these variables. The dissimilarities between the correlations presented in Table 1 motivate this work. They highlight the challenge of studying convective organization and they show that different indices may consider different aspects of convective organization.



## 3  Analysis strategy

### 3.1  Conditions to be satisfied

The studied indices were designed with the purpose of quantifying the degree of deep convective organization. Estimating their skill is challenging because organization has no formal definition. However, we can assess if the indices satisfy some expected conditions of convective organization. We have chosen seven conditions, which can be grouped into three categories and will be evaluated via sensitivity studies.

The first category concerns the behavior of the indices towards small perturbations. The indices should not significantly change

1. when one random grid box in the domain is set to convective,

2. when two objects merge into one by adding one single grid box as convective.

Both conditions are meant to study the effect of noise. To simulate the difference in behavior, we randomly add in the first case one grid box in a non-convective region of the image, thus the number of objects increases by one. In the second case, we add the convective grid box between two close objects in such a way that the objects merge and the number of objects decreases by one. The indices that satisfy the criteria above are called noise-safe.

The second category evaluates the intrinsic behavior of the indices. The indices should:

3. decrease when objects are moving apart,

4. increase when one object's size is increasing.

Condition (3) is the most relevant one of all because it defines the relationship between the proximity of the objects and the organization. The indices that do not satisfy it shall not be used to quantify organization. Condition (4) states that the organization gets stronger when the area of the objects is increasing. This condition is not explicitly taken into account in all convective indices. ROME, COP, ABCOP, MICA are built with this assumption, while $I_{org}$, $L_{org}$, SCAI, and MCAI do not consider the area of convective objects in their formulation. If this assumption is not satisfied, one potential approach to study organization is to stratify the images by the total area of convection. This option is marginally performed by Tobin et al. (2012) where the values of organization are compared within similar intervals of total precipitation. Either way, if condition (4) will prove to be important or if it will be considered to be irrelevant, we evaluate it for all the indices in order to provide awareness of the relationship between the indices and the area of convection.

The third category evaluates the dependency on the resolution and on the domain limits. This set of conditions is crucial to compare conclusions from analyses of organization using different data. The indices should not significantly change:

5. when using a slightly different spatial resolution,

6. when using data taken at a slightly different time,

7. when choosing a slightly different spatial domain.



## 3.2 Comparison strategy

In order to quantify the dependence of the indices on the specific perturbance under consideration, we modify the images accordingly: for each image, we compute the value $i$ of each index (reference), then we modify the image as required by the condition under study, and we compute the new value of the index $i'$ (perturbation). The difference $i' - i$ shows how much the value of the index has changed. However, the organization strength is given by ranking the value $i$: for example, extremes are defined by percentiles. Therefore, instead of comparing the absolute change in the values of the indices $i$, we compare their percentiles $p(i)$. Let $f(x)$ be the measured distribution function of a certain index, the percentile of the value $i$ is defined by:

$$p(i) = 100 \int_{-\infty}^{i} f(x) dx \tag{1}$$

where the factor 100 is set to have values from 0 to 100. A value of $p(i) = 90$ means that the convection is in the 10% most organized disposition.

Using percentiles is particularly advantageous because the difference $\Delta p = p(i') - p(i)$ really quantifies the amount of change in organization estimated by the indices. For example, a difference $\Delta p = 10$ means that the perturbation results in a jump of the organization of 10% of the events. Since $\Delta p$ is dimensionless, it can be compared between different indices. Consequently, we quantify changes in each index with $\Delta p$. In the following, an average value of $\Delta p < 3$ is considered small, while an average value of $\Delta p > 10$ is considered large.

We have found that it is very difficult for a single index to satisfy all the presented conditions. In particular, as shown in the next section, none of the considered indices does that. Hence, we developed a new index, called OIDRA.

## 4 Results

### 4.1 First category: noise-safeness

In this section, we study conditions (1) and (2). Condition (1) states that the consequences on the indices must be small when one grid box is added as a new convective object. The consequences of adding a single convective grid box are also shown by Retsch et al. (2020) and Jin et al. (2022). However, only a few cases have been examined. Instead, this work provides a statistical picture that validates previous results, proving that those cases were not cherry-picked.

Condition (2) states that if there are two objects as close as one grid box, merging them does not produce significant changes in the value of the indices. The validity of this condition stems from the fact that the presence of an additional grid box does not substantially alter the disposition of convection. Consequently, it is expected that the inclusion of this grid box would not significantly affect the values of the organization indices. Condition (2) has never been discussed in studies of organization. Still, it is of great importance because when two objects merge, the sizes, number, and distances between objects change, thus changing the value of the indices. Therefore, both conditions (1) and (2) contribute to comprehending the potential impacts of any source of noise on the organization indices.



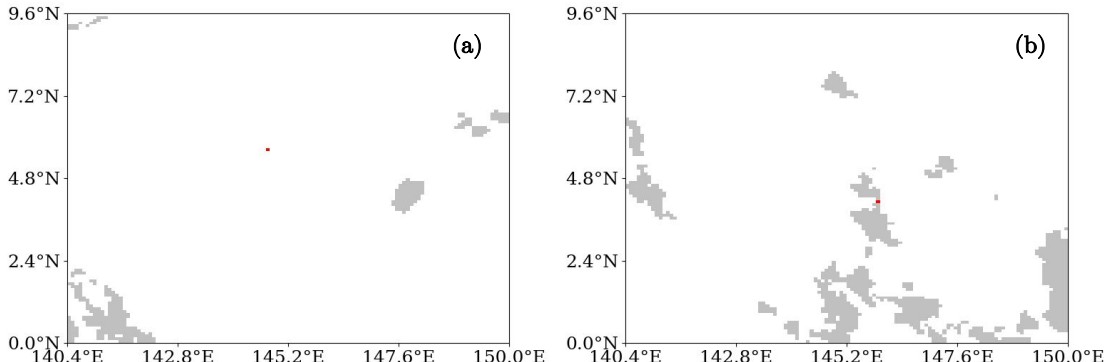

**Figure 2.** (a) Convective objects in the domain of interest are shown in grey for the 4th of January 2012 at 12:30 UTC, the additional convective grid box is in red and it shows the perturbation associated with condition (1). (b) Convective objects in the domain of interest for the 1st of January 2012 at 21:00 UTC, the additional grid box is in red and it shows the perturbation associated with condition (2).

Fig.2 illustrates the perturbations implemented for the two conditions. For the sensitivity study (1), the additional grid box is always placed randomly and it is never in contact with other objects. For the study (2), the additional grid box always merges two objects. The perturbation shown in Fig.2b is possible only when there are two objects as close as 1 grid box. Therefore, only those events are used to evaluate condition (2) (26255 events out of 76462).

The procedure is similar for all sensitivity studies: for each image, the indices of organization are computed (labeled as 'reference'), then the associated perturbation is applied and the indices are computed on the new image. The set of images after the perturbation is labeled as 'perturbed'. As an illustration, Fig.3a and Fig.3b compare the distributions of $I_{org}$ and COP before and after the perturbation of condition (1) using the complete statistics. As expected, the random noise is producing on average a decrease in both $I_{org}$ and COP. Though, for $I_{org}$, a non-negligible fraction of events close to zero is redistributed at higher values. Those events are mainly images with only two objects that are placed at opposite sides of the image (not shown). Therefore, the additional grid box reduces the nearest neighbor distances and increases the value of $I_{org}$. Fig.3c and Fig.3d show the distributions of the percentiles for $I_{org}$ and for COP. The percentiles are always computed with respect to the reference distribution, thus the distributions of the reference dataset are flat while the ones of the perturbed dataset are not.

The grey and red arrows indicate the values of $I_{org}$ and COP for the event shown in Fig.2a. The value of $I_{org}$ and COP move from 0.80 to 0.70, and from 0.32 to 0.28 respectively. The corresponding percentiles move from 0.91 and 0.78 and from 0.14 to 0.06, leading to differences of $\Delta p(I_{org}) = 13$ and $\Delta p(COP) = 8$. This means that one additional grid box, even if it is just one out of $120^2$, changed the rank of the value of $I_{org}$ by 13% and the rank of COP by 8%. It is also worth noting that the case depicted in Fig.2a is classified as highly organized according to $I_{org}$, whereas it is classified as highly disorganized according to COP.

The distributions in Fig.3e and Fig.3f present the relationship between the percentiles of the reference and of the perturbed dataset for $I_{org}$ and for COP. The two figures are similar and they show that both indices decrease on average. Thus the new





**Figure 3.** Distribution of $I_{org}$ (a) and COP (b) for both the reference and the perturbed dataset. Percentile distribution of $I_{org}$ (c) and COP (d). Bidimensional distribution of $p(I_{org})$ (e) and $p(\text{COP})$ (f) of the reference and the perturbed dataset. The red and the grey arrows indicate values and percentiles of $I_{org}$ and COP for the configuration in Fig.2a respectively with and without the additional grid box.

convection dispositions are correctly identified as less organized. Moreover, the vertical spread of the distribution comes from




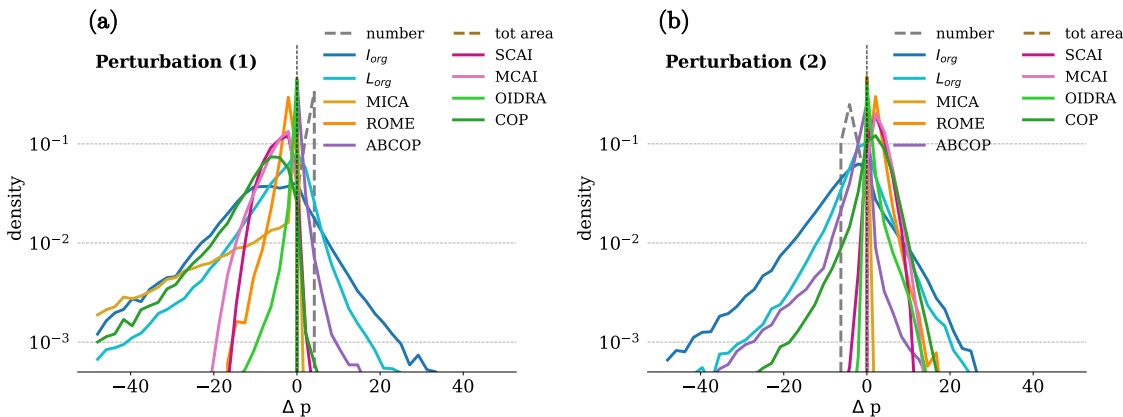

**Figure 4.** Distribution of $\Delta p$ for all the indices (a) for condition (1), and (b) for condition (2).

the randomness of the additional grid box position. Fig.3 proves that the behaviors of $I_{org}$ and COP under noise are similar even if their reference distributions are very different.

The results of the sensitivity study for all the indices and for both conditions are shown in Fig.4 as the distributions of $\Delta p$. The difference $\Delta p$ produced by the perturbation of condition (1) has a long tail for $I_{org}$, $L_{org}$, COP, and MICA. A long tail means that one single grid box can heavily modify the index value. The other indices are more peaked at zero. For SCAI and MCAI, the $\Delta p$ distributions derive mostly from the average number of objects. In the other cases, the enhanced peak at zero occurs because the indices are closely related to the total area of the objects in the domain, and thus adding one single grid box

does not affect their value significantly. Under perturbation (2), all the indices peak at zero, proving that merging two close objects does not affect their value much. However, $I_{org}$, $L_{org}$, COP, and ABCOP show a significant tail at negative $\Delta p$.

     The most frequent sign of $\Delta p$ is very important because it indicates if the indices generally classify the perturbed dataset as more or less organized. In most cases, the sign is negative for perturbation (1), meaning that the organization is reduced. The only index that classifies the perturbed dataset as more organized is ABCOP, which is in contrast with the other organization

indices. In particular, adding one convective grid box always produces an increase in the value of ABCOP. For perturbation (2), $I_{org}$, $L_{org}$ and ABCOP predict a decrease of organization, while SCAI, MCAI, COP, ROME, MICA, and OIDRA predict an increase after the perturbation.

     In order to quantify the average change of the indices values, we report in Table 2 the average of the absolute percentile difference $<|\Delta p|>$ for each index under study. Large values reveal a higher sensitivity to noise, while low values indicate

noise-safe indices. The indices ABCOP and OIDRA do not change significantly under the perturbation (1). Under the perturbation (2), the indices MICA and OIDRA do not change significantly, while $I_{org}$, has large changes.

     For all indices and for both conditions, $\Delta p$ is influenced by the average and distribution of the number of objects of the reference dataset: images with a larger number of objects show less sensitivity to noise. Therefore we report the values of $<|\Delta p|>$ as a function of the number of objects in the reference dataset in Fig.5. For both perturbations, the sensitivity of all





**Table 2.** Average of the absolute change in percentile $< |\Delta p| >$ after the perturbations of condition (1) and condition (2).

|  | $I_{org}$ | $L_{org}$ | SCAI | MCAI | COP | ABCOP | ROME | MICA | OIDRA | number | total area |
|---|---|---|---|---|---|---|---|---|---|---|---|
| condition (1) | 11.7 | 7.8 | 4.9 | 5.2 | 10.4 | 0.6 | 2.5 | 6.7 | 0.4 | 3.5 | 0.0 |
| condition (2) | 8.7 | 5.5 | 2.8 | 3.3 | 4.2 | 2.8 | 2.6 | 0.0 | 0.8 | 3.9 | 0.0 |

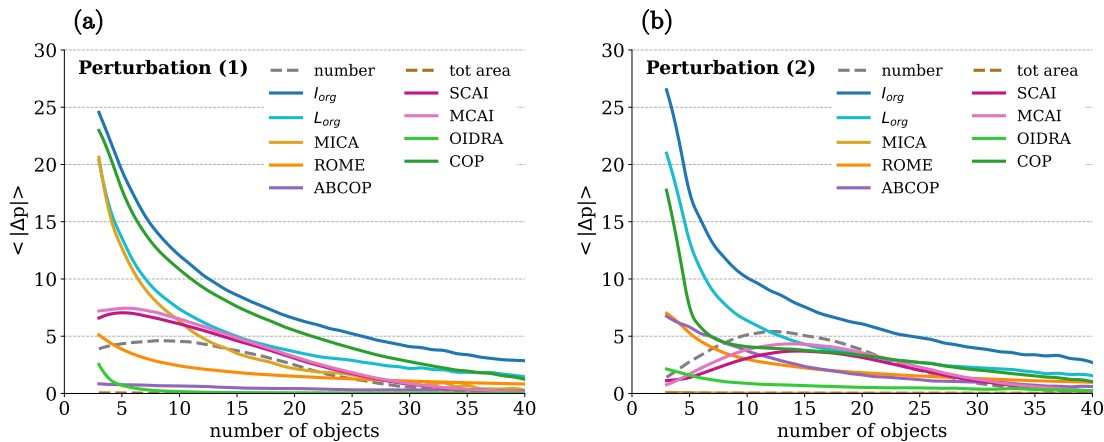

**Figure 5.** Senstivity of each index as a function of the number of objects of the reference dataset. Perturbation (1) is shown in (a) and perturbation (2) is shown in (b).

indices decreases as the number of objects increases. As a result, each index becomes noise-safe when a sufficient number of objects is present. The index $I_{org}$ is very sensible to noise at low number of objects, and it becomes noise-safe when more than 35 objects are present. The index $L_{org}$ has a similar behavior as $I_{org}$ for a low number of objects, but it becomes more robust to noise as the number of objects increases. The sensitivities to noise of SCAI and MCAI follow the distribution of the number of events shown in Fig.5 because of the high correlation between them. Generally, the indices that take into account the area of the objects become noise-safe more rapidly than the ones that do not. Among the various methods available to quantify convective organization, ROME, OIDRA, and the total convective area are demonstrated to be the most robust against noise.

## 4.2 Second category: intrinsic behavior

In this section, we study conditions (3) and (4). Condition (3) states that the degree of organization must increase with the proximity of the objects. This condition is the most important one, and no index of organization can miss it because it encapsulates the essence of convective organization. Condition (4) states that the degree of organization must increase when the object sizes are increasing. The sensitivity study associated with this condition presents distinctly the role of the object sizes for each index.



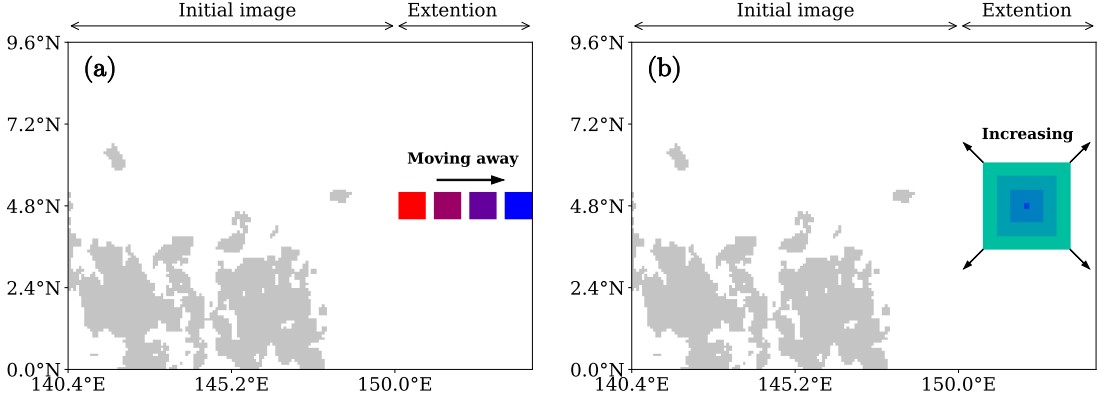

**Figure 6.** Convective objects in the domain of interest similarly to Fig.2. Perturbation of conditions (3) in (a), and (4) in (b) are shown for the 1[st] of January 2012 at 13:30 UTC. The leftmost part shows the initial image. The rightmost part shows the extension of 50 grid boxes, where a test object is added. In (a) the extra object has a size of 10x10 and it is shifted rightward. In (b) the extra object is centered and its size is increased.

The procedure is similar for conditions (3) and (4). Firstly, every image in the dataset has been extended rightward by 50 grid boxes with an empty space. Then an object, called 'test object', is placed in the extension. For condition (3), the test object

is as big as 10x10, it is placed in the leftmost part of the extension, and the perturbation consists in moving it rightward up to the end as shown in Fig.6a. The perturbation is parametrized by the magnitude of the displacement. For condition (4), the test object has a squared shape of 2x2, it is placed in the center of the extension, and the perturbation consists in increasing its size up to 48x48 as shown in Fig.6b. The perturbation is parametrized by half the length of the object side. The reference configuration is the one where the test object is present and is not perturbed, and all the other configurations are compared to

that one.

Fig.7a and Fig.7b show the distribution of $\Delta p(\text{COP})$ as a function of the perturbation of both conditions. The two figures show that COP has the correct trends for both perturbations: COP decreases when the test object is moving away, and it increases when the test object size is increasing. In addition, the high correlation with the reference dataset proves that the value of COP is still linked to the reference value after the perturbations. The mean $<\Delta p>$ is reported in Fig.7c and Fig.7d

for all the indices as a function of the perturbation of both conditions. All the trends are negative, thus all indices correctly increase with the proximity of the objects. However, the trends have different magnitudes: ROME and ABCOP have very little dependency on the test object position. Oppositely, MICA shows a very large dependency on the object position. Such a large value is due to the specific definition of MICA (see Appendix A0.8) and to the strategy used to evaluate condition (3): the perturbation affects $A_{cld}$, thus it proportionally affects the value of MICA. The index $I_{org}$ exhibits a rapid decrease initially,

followed by stabilization because it is not sensible to organization beyond the $\beta$-mesoscale. This behavior has been corrected in $L_{org}$, which looks to be sensitive to the object's position across all distances. Even if this condition should be satisfied by construction, it has never been validated before this work, and it is always taken for granted.





Fig.7d displays the behavior of the indices when the test object's size is increasing. Firstly, SCAI, $I_{org}$, and $L_{org}$ are constantly zero because their values do not depend at all on the object sizes. COP and MCAI increase linearly, while ABCOP

and ROME have increasing trends that are not linear nor quadratic. Specifically, ROME has a quadratic increase in its value (not the percentile). Similarly, MICA has a linear increase in the original value (not shown). ABCOP exhibits low sensitivity to the test object size when it is small. However, as the length scale increases, the response correspondingly amplifies. Lastly, OIDRA first decreases and then increases with respect to the object's side. The decrease is due to the effect of the total area $\sum_i A_i$ at the denominator in the definition of OIDRA. Except for SCAI, $I_{org}$, and OIDRA, all indices predict higher degrees of

organization for a larger test object, thus they represent bigger objects as more organized. Nevertheless, OIRDA is correlated with the mean object size with a correlation coefficient of 0.51 (Table 1).

### 4.3 Third category: Capacity to compare across diverse datasets.

Having multiple independent studies that agree with each other is crucial for advancing scientific understanding. It shows that the results are reliable and can be trusted. In the specific case of convective organization, those independent studies

can be performed by analyses that use different datasets. Therefore, it is crucial to ensure that the indices of organization do not depend significantly on the dataset characteristics. Otherwise, achieving consensus among different studies becomes exceedingly challenging. Conditions (5), (6), and (7) are meant to evaluate the possible dependency of the indices on the resolution and domain limits. In the following, each of them is discussed separately.

### 4.3.1 Condition (5): sensitivity to the horizontal spatial resolution

The datasets that are used to study convective organization may be different, in particular, the horizontal resolution used in the various analyses may vary up to one order of magnitude. It is thus of great importance to understand the role of horizontal resolution on the organization indices. In this section, we evaluate condition (5) which states that the horizontal resolution of the dataset should not influence significantly the value of the organization indices. The role of the resolution is studied by down-scaling each image of factors 2, 3, 4, 5, and 6. In other words, the images, that are initially 120x120, are reduced to

60x60, 40x40, 30x30, 25x25, and 20x20 grid boxes. The reduction is performed by grouping the initial images' grid boxes in blocks of $nxn$, then the mean values over the blocks are computed where the values 1 and 0 are associated with convective and non-convective regions. If the mean value is larger than 0.5, the block is set to 1 (i.e. it is associated with convection), otherwise, if the mean value is smaller or equal to 0.5, the block is set to zero (it is not associated with convection). An example is given in Fig.8, comparing an image with the original resolution of 120x120 grid boxes and an image with the resolution down-scaled

by a factor 3, so that it consists of 40x40 grid boxes. The set of images with the original resolution is considered to be the reference sample, and the ones with different resolutions are the perturbed datasets.

When the grid box size is rescaled, the value of each index may change correspondingly, depending on the index units. ROME and the total area of convection have a dimension of km$^2$, therefore the area per grid box has to be corrected when the resolution is changed. SCAI and MCAI are inversely proportional to the number of grid boxes, thus they quadratically increase

with the resolution down-scaling. All the other indices are dimensionless and they do not have any explicit dependency on the



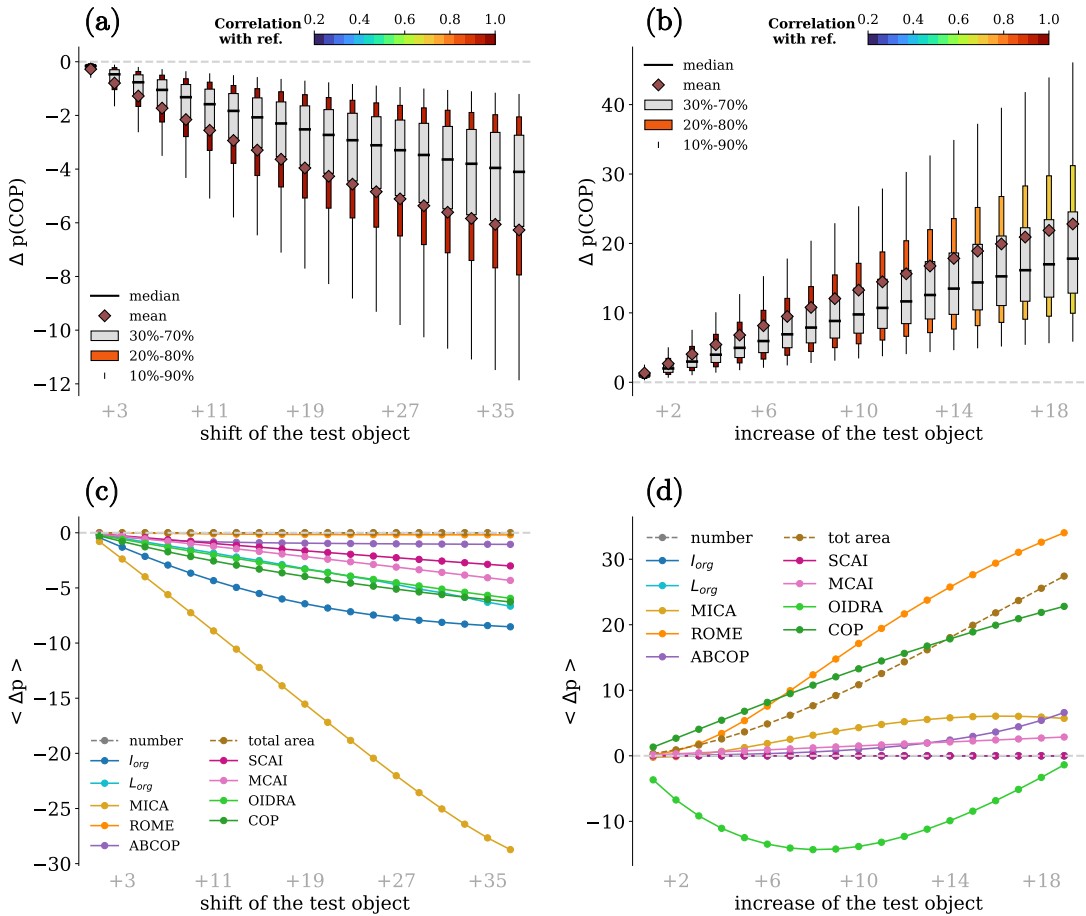

**Figure 7.** Distribution of $\Delta p(\mathrm{COP})$ as a function of the perturbation (3) in (a), and perturbation (4) in (b). The grey boxes indicate the percentile ranges from 30% to 70%, the colored boxes indicate the percentile 20% and 80% and the colors display the correlation between the reference and the modified dataset. The whiskers cover from 10% to 90% of the distributions. The means and the medians of $\Delta p(\mathrm{COP})$ are shown by the rhombuses and the black lines respectively. The means of $\Delta p$ are displayed for all the indices as a function of the perturbations of condition (3) in (c), and condition (4) in (d).

resolution. These different scaling factors are removed by multiplying the indices by the inverse factor so that, in principle, the average value of the indices should not change. Such an operation is crucial to study differences between the reference and the modified dataset.

Fig.9 shows the distribution of $\Delta p$ for all the indices under a change in resolution of a factor of 3. It is noteworthy that
ROME, SCAI, and MCAI have a clear peak at zero, proving that the artificial scaling described above is correct and that we understand the resolution's dependency on the indices well. The distributions of $\Delta p$ exhibit a generally broad range, indicating the significant impact of the resolution on the values of the indices. The most important effect of down-scaling the resolution is

 

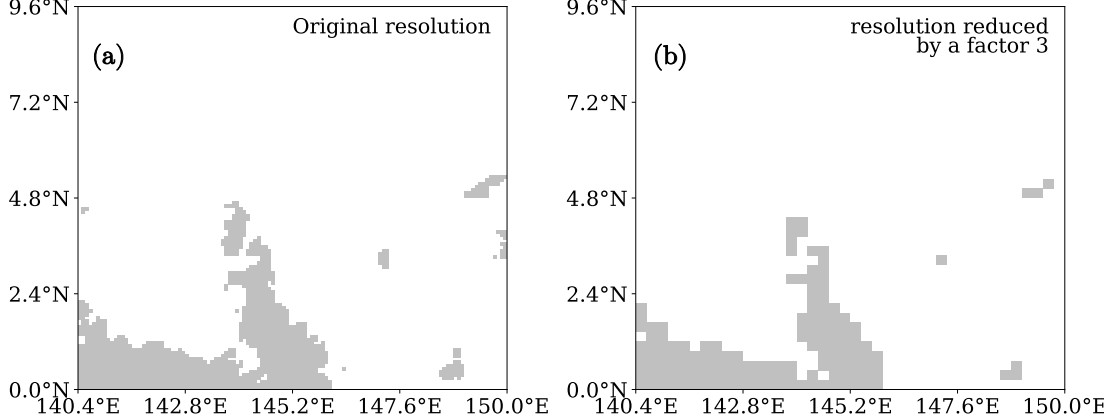

**Figure 8.** Convection in the domain of interest on the 1[st] of January 2012 at 02:00 UTC with the original resolution (a) and with a 3 times worst resolution (b).

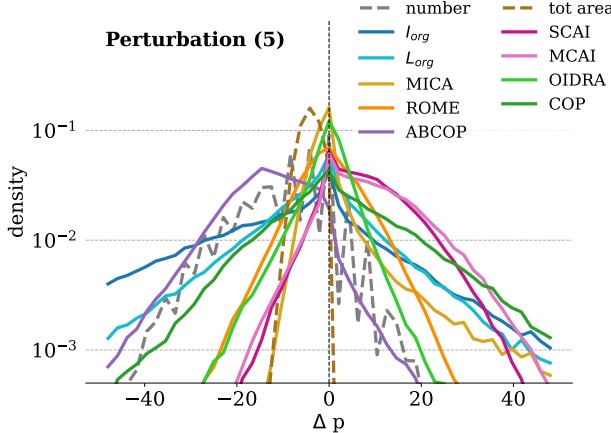

**Figure 9.** Distribution of $\Delta p$ for all the indices after down-scaling the resolution of a factor of 3.

due to the smaller objects in the domain (the ones with sizes of 1 or 2 grid boxes). The smaller objects can be lost (if they are isolated) or can be merged with bigger objects nearby, and in both cases, the consequences can be significant. These effects are
particularly pronounced for $I_{org}$: the disappearance of the small objects produces an increase in the nearest neighbor distances and consequently, a decrease in the value of $I_{org}$. Oppositely, COP is influenced by the size of the objects, thus its value is increasing with lower resolutions. The distributions of $I_{org}$ and COP are shown in Fig.10 as a function of the resolution down-scaling factor as described in Sect.4.2. When the resolution reduction does not change the reconstructed number of objects the value of $\Delta p$ is close to zero for all the indices.



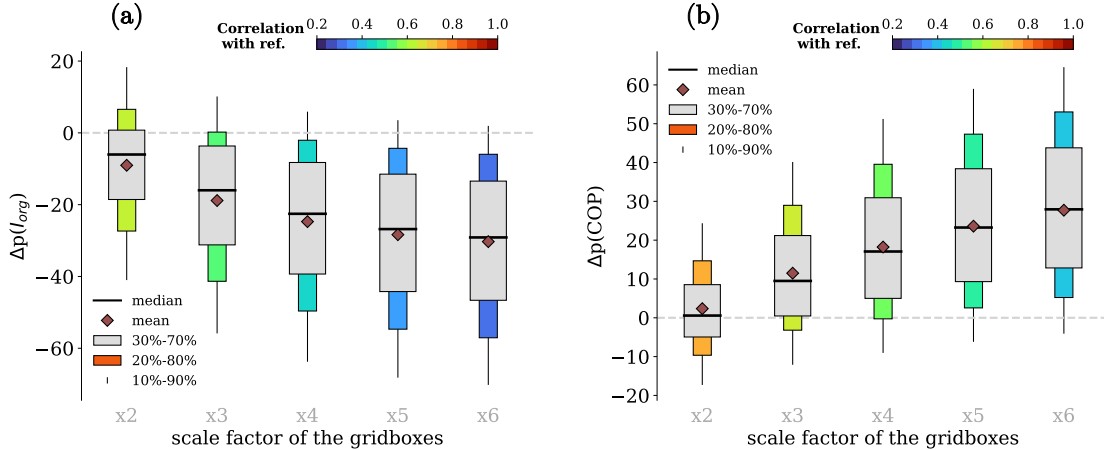

**Figure 10.** Distribution of $\Delta p(I_{org})$ (a) and $\Delta p(\text{COP})$ (b) as a function of the resolution scale factor. Description is similar to Fig.7.

The mean of $\Delta p$ is shown in Fig.11a for different resolutions for the indices. When the resolution scale factor is small, the values of $<\Delta p>$ are close to zero, and they move away from zero when the scale factor is increasing. The sign of $<\Delta p>$ is different for different indices: it is negative for ABCOP and $I_{org}$, and it is positive for all other indices. The index which shows less sensitivity to different resolutions on both spread (not shown) and average of $\Delta p$ is OIDRA.

The most important information for comparing the datasets with different spatial resolutions is given in Fig.11b. The figure shows the correlations between the reference dataset and the datasets with lower resolutions. They vary with respect to the originally observed values, decreasing monotonically with increasing scaling factor. The total area is not much affected by the change in resolution, and, along with it, the indices more related to the total area are also the ones less sensible to changes in resolution. The indices which are more affected are $I_{org}$, $L_{org}$, and COP. The correlation of $I_{org}$ drops to 0.5 when the resolution is down-scaled by a factor of 3. The index which has a higher correlation between different resolutions is OIDRA.

The presented results are of great importance for comparing the results of different studies. Different analyses often have different resolutions, thus, they can have discrepancies even if they make use of the same organization index. This effect has to be summed up with other sources of uncertainties, like noise or different thresholds as remarked in Sect. 4.1, and the resulting discrepancies can be considerable. In particular, they can be very important when $I_{org}$ is used, and they can lead to different results even in similar analyses.

### 4.3.2 Condition (6): sensitivity to observation time

The time variability of the organization index has been already shown (Tompkins and Semie, 2017; Cronin and Wing, 2017; Muller and Romps, 2018; Muller et al., 2022), and seasonal and diurnal cycles of organization have been studied using satellite observations (White et al., 2018; Stubenrauch et al., 2023). In most analyses, convective organization indices are computed per snapshot and then averaged over long periods, while small variability is not examined. In this section, we analyze the




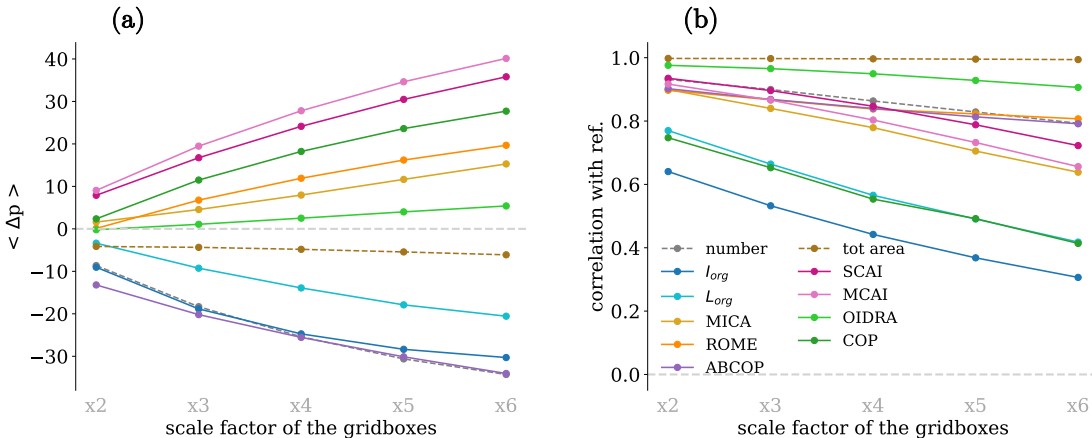

**Figure 11.** Means of $\Delta p$ (a) and correlations with the reference (b) are displayed for each index as a function of the resolution scale factor.

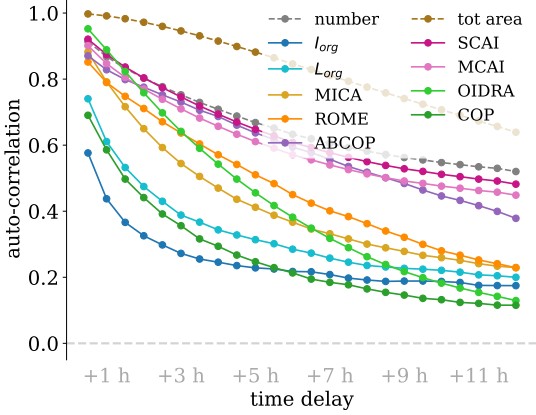

**Figure 12.** Autocorrelation of each index between 30 minutes and 12 hours.

variability of the indices on a small time scale, evaluating condition (6). In principle, the daily cycle can affect all the results of this analysis because we are comparing the indices at different times. Therefore, the effect of the daily cycle has been calculated. Since it is found to be negligible for the considered region, its effect is not shown.

    The variability with time for each index is summarized in Fig.12, which shows the autocorrelation of the indices as a function of the time shifts from 30 minutes to 12 hours. The autocorrelations have large values for small differences in time, while they

decrease with increasing shifts. There are large differences between indices: COP, $I_{org}$, and $L_{org}$ have a very rapid decrease, with correlations already lower than 0.5 for COP and $L_{org}$ and 0.4 for $I_{org}$ after one hour. This means that these indices cannot be compared when computed with more than one hour distance and that they cannot be related to any physical variable which





**Table 3.** Average of the absolute change in percentile $<|\Delta p|>$ due to different shifts in time. The last row shows $<|\Delta p|>$ for a random shuffle of the dataset.

| | $I_{org}$ | SCAI | MCAI | COP | ABCOP | ROME | MICA | OIDRA | number | total area |
|---|---|---|---|---|---|---|---|---|---|---|
| $<|\Delta p(|\Delta t = 30 \text{ min})|>$ | 18.5 | 7.6 | 8.6 | 13.4 | 7.9 | 6.7 | 4.3 | 5.4 | 7.9 | 1.5 |
| $<|\Delta p(|\Delta t = 90 \text{ min})|>$ | 24.4 | 11.4 | 12.6 | 19.1 | 10.5 | 11.1 | 9.6 | 11.6 | 11.3 | 3.9 |
| $<|\Delta p(|\Delta t = 12 \text{ hours})|>$ | 30.8 | 21.8 | 22.6 | 29.8 | 22.6 | 26.2 | 23.4 | 30.6 | 20.7 | 18.0 |
| $<|\Delta p(|\Delta t = 6 \text{ months})|>$ | 33.4 | 33.3 | 33.3 | 33.3 | 33.4 | 33.4 | 30.1 | 33.4 | 32.8 | 33.1 |

has different timescales. The autocorrelation of ROME, MICA, and OIDRA is close to 1 for a small time, while it decreases toward zero with time scales of a few hours. Lastly, the autocorrelation of SCAI, MCAI, and ABCOP has a slower decrease,
and it is still about 0.5 after 12 hours. All the autocorrelations reach zero after a few days.

In order to further explore the indices fluctuations with time, the differences $<|\Delta p|>$ are given in Table 3 for a time shift of 30 minutes, 90 minutes, 12 hours, and 6 months. The table shows that $<|\Delta p|>$ is increasing with time for all the indices and that the increase rate can differ depending on the specific index. The differences $<|\Delta p|>$ also reflect the autocorrelation values and trend: where the autocorrelation is lower, $<|\Delta p|>$ is larger. For very large time shifts, the autocorrelation is zero,
and $<|\Delta p|>$ moves towards the limit 33.3 (not shown), the exact value does not depend on the atmospheric dynamics, but it depends on the amplitude of the seasonal cycle, the daily cycle, and on if there are any points of accumulation in the distribution of the indices. The values of $<|\Delta p|>$ are very large for COP, $I_{org}$, and $L_{org}$, and they already are larger than 10 when the time shift is 90 minutes (except for MICA which is 9.6). This means that the organization ranking given by indices is moving on average by more than 10% every 90 minutes. After a time shift of 12 hours, the differences $<|\Delta p|>$ are larger than 20, and
in particular, they are close to 33.3 for COP, $I_{org}$, $L_{org}$, and OIDRA in agreement with the low correlation shown in Fig.11.

This analysis can be of help in two cases. Firstly, when two studies employ the same organization index but compute it with a time lag, the two indices may exhibit a weak correlation, making direct comparisons challenging. Secondly, when organization is linked to other atmospheric variables that are measured with a time delay (e.g., from a different instrument), the rapid variability of organization may cause the relationship to weaken or disappear due to the time lag. By considering these
cases, this analysis helps to account for temporal differences and enables a more accurate understanding of the relationship between organization and other variables in atmospheric studies.

### 4.3.3 Condition (7): sensitivity to the domain limits

Convective organization is computed from the disposition of convection in a specific domain. Different studies may not target exactly the same domain, leading to possible differences in their final results. In this section, we analyze condition (7), which
states that organization indices must not be significantly different on similar domains. Especially, we analyze the changes of the indices of organization for domains that overlap from 80% to 99%. Such a study is performed by taking into consideration the region 0.8°N-8.8°N x 140.4°E-148.4°E made by 100x100 grid boxes with 0.08° resolution. This region is shifted eastward





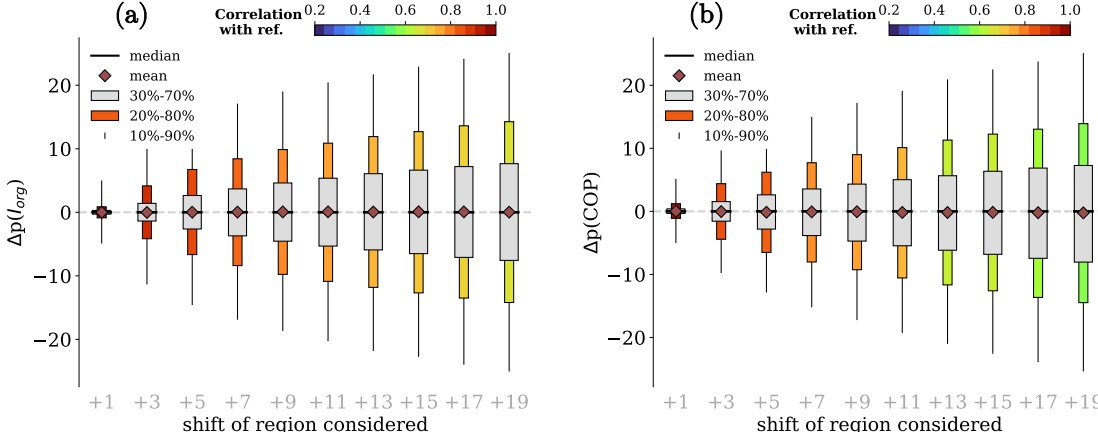

**Figure 13.** Distribution of $\Delta p(I_{org})$ (a) and $\Delta p(\text{COP})$ (b) as a function of the perturbations of condition (7). Description is similar to Fig.7.

one grid box at a time, and for each shift, the indices of organization are computed. The shift of 1 grid box eastward changes the domain by 1%.

Fig.13 shows the distributions of $\Delta p$ for $I_{org}$ and for COP as a function of the eastward shift of the domain. The distributions of $\Delta p$ are very narrow for small shifts, and they get broader with larger shifts. The mean and the median are very close to zero with no trends, and the correlation decreases with the magnitude of the shift. The variability of both $\Delta p(I_{org})$ and $\Delta p(\text{COP})$ is similar, as well as the correlation with the reference dataset.

The correlations of the indices between the shifted and the initial domains are shown in Fig.14. They are large for small

shifts and they are slowly decreasing as the domain moves eastward. In particular, for small shifts, the trend is about 0.01 per grid box, and since a one-grid box shift corresponds to 1% of the image, the correlations are decreasing approximately at a rate of 0.01/%. For large shifts, the indices are divided into two groups: SCAI, MCAI, ROME, and ABCOP decrease slowly, while COP, OIDRA, MICA, and $I_{org}$ show a more rapid decrease.

The results obtained with this analysis prove that the indices of organization have small differences when similar domains

are considered. The larger the differences between domains, the larger the discrepancies of the indices are. Thus, all the indices show a small rate of change under the shifting of the domain, proving that all the indices satisfy condition (7).

## 5    Conclusions

Convective organization is an emerging topic that has been receiving a lot of attention in recent years because of its potential implications for weather and climate. Several indices have been developed to identify organization and they became an

essential tool to deepen our knowledge on this topic. In this study, we assess the reliability of each index by employing inno-





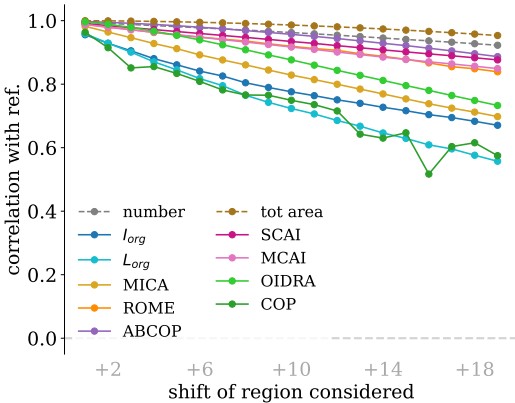

**Figure 14.** Correlation of each index between the reference and the modified datasets as a function of the domain shift.

vative methodology. Initially, we established a set of criteria that are anticipated for convective organization. Subsequently, we evaluated the indices to ascertain whether they met these established conditions.

In this article, we compared 9 object-based indices. The conditions are assessed by applying different perturbations to the dataset and measuring the repercussion on the indices. The results are obtained using the TOOCAN systems mask. The entire

analysis is run also using the generated dataset shown in Fig.1. The results are similar and they are shown in the supplement material.

In the previous sections, the indices are compared for each condition. In the following list, we summarize the results for each index, combining the different conditions to obtain the global picture that emerges from the results:

– $I_{org}$: The index $I_{org}$ proves to be inadequate for quantifying convective organization, both in terms of consistency and

in terms of comparability. First of all, it is very sensitive to noise. Secondly, when two close objects are merged, $I_{org}$ has large changes, and on average, it predicts the new configuration to be less organized. In particular, the index $I_{org}$ is not noise-safe when the number of objects in the domain is less than 35. Furthermore, $I_{org}$ is totally insensible to the objects' sizes. Another weakness of $I_{org}$ is its large sensitivity to the images' resolution. In fact, identical images with different resolutions produce significantly different values of $I_{org}$, the differences are in both the mean values and

the distributions. Lastly, the time auto-correlation is decreasing rapidly, thus the relationship with other atmospheric variables may be lost if it is not searched exactly at the right time.

– $L_{org}$: The behavior of $L_{org}$ is similar to the one of $I_{org}$. However, it has better behaviors for all the studied perturbations. First, it is more robust to noise than $I_{org}$: the index $L_{org}$ becomes noise-safe when there are more than 20 objects in the domain. Second, it is less sensible than $I_{org}$ to the spatial and temporal resolutions, and the domain limits. Moreover,

$L_{org}$ shows a dependency on the object's position across all distances, confirming its sensitivity to organization beyond the $\beta$-mesoscale. Finally, both $I_{org}$ and $L_{org}$ are not sensitive to the object's size.



- COP: The index COP, similarly to $I_{org}$, is sensitive to noise, and it becomes noise-safe when more than 25 objects are present in the domain. It correctly increases with the proximity of objects and it correctly increases with the objects' size. On the other hand, COP is sensitive to the images' resolution. In particular, the same image with the original resolution and three times worse resolution gives values of COP that have a correlation of 0.65. For this reason, COP is not a good index for comparing different studies. Lastly, similarly to $L_{org}$, COP varies very rapidly with time.

- ABCOP: The index ABCOP has a large correlation with the total objects' area $\sum_i A_i$ as shown in Table 1. This relationship is uniquely coming from the larger objects in the images, while the smaller objects do not play a significant role in the value of ABCOP. This behavior comes from the $\max_{j \neq i}()$ function in the ABCOP definition (equation A8), which gives great importance to large objects. As a consequence, ABCOP is one of the most noise-safe indices. The index ABCOP is also very dependent on the number of objects because it is defined as a sum over each object instead of as a mean like COP. This characteristic is visible in three of the studied conditions. First, when one convective grid box is added randomly in the domain, ABCOP increases instead of decreasing because the number of objects is increasing. Second, when two close objects are merged, ABCOP decreases instead of increasing because the number of objects decreases. Third, when images with different resolutions are analyzed, the differences in ABCOP follow the differences in the number of objects. However, even if there are large differences, ABCOPs measured on different resolutions are highly correlated. ABCOP shows a similar behavior when it is computed on slightly different domains or after a small time lag. Therefore, ABCOP is a good index to compare results obtained with different instruments. Last but not least, the index ABCOP is correctly increasing with the proximity of the objects and the increment is relatively lower than the ones of the majority of indices.

- SCAI and MCAI: The indices SCAI and MCAI have similar behaviors. Their value is highly correlated with the number of objects (R=-0.96[1]), therefore their behaviors reflect the one of the number of objects. Upon adding a single random convective grid box to the images, both SCAI and MCAI show a decrease in their value. Furthermore, these indices indicate an increased level of organization when two nearby objects are merged. Both indices correctly exhibit an increase in value as the proximity of the object increases, with MCAI being slightly more sensitive than SCAI. The main distinction between these two indices lies in their dependence on the object's size. SCAI displays no sensitivity to object size, whereas MCAI takes into account the object's size, thereby rectifying SCAI's wrong behavior. Changes in resolution have an impact on the values of SCAI and MCAI, primarily due to variations in the number of objects. Nonetheless, their correlations with the original resolution decrease slowly, making them suitable indices for comparing different resolutions. Similarly, SCAI and MCAI are the indices that are less affected by shifts in time and in space.

- ROME: The index ROME satisfies most of the conditions presented here, however, its biggest problem is the very high correlation with the mean objects' size $\sum_i A_i/N$ as shown in Table 1. Thus, it is not convenient to quantify convection with ROME (equation A9) when a simpler formula can be used to obtain a very similar value. ROME is noise-safe enough, having values of $\Delta p$ smaller than 3. Knowing the correlation with the mean area, we can state that those

---

[1]We recall that SCAI and MCAI are negated in this work, hence the correlation with the number of objects is negative.





conditions are satisfied because of the dataset instead of the index behavior. Among all, ROME is the index that depends less on the proximity of the objects, with a variation of only $\Delta p \lesssim 0.3$ when a 10x10 test object is moving away from the other convective objects. Moreover, ROME depends very much more on the size of the objects. The dataset's horizontal resolution does not strongly affect the ROME index, and the autocorrelation shows a slow decrease with time. The changes of ROME with the considered domain are about 0.5% for a 1% shift of the domain.

– MICA: The index MICA is particular because of its unique definition which has to be kept in mind when reading these results and when using MICA for performing an analysis. The obtained results here are very much influenced by the type of data and the adopted strategy. For example, this work suggests that MICA is greatly influenced by the proximity of objects, however, this is not true in general, and it is true only when the test object is changing $A_{cld}$ (defined in Appendix A0.8). Similarly, the test object size studied in Sect. 4.2 is also influenced by the changing in $A_{cld}$. One other example

is the result of Sect. 4.1, where MICA predicts a very small increase when two close objects are merged. The simulated noise produces a long tail of $\Delta p$ which means that a source of noise can produce a large change. The reason is that MICA does not take into account each object size, and it strongly depends on the disposition of the objects. Thus, if a simulated noise is placed far from the convective objects the value of MICA is decreasing significantly. For the same reason, MICA is increasing when the resolution is reduced because small objects may be lost, reducing the area $A_{cld}$.

The auto-correlation of MICA is decreasing rapidly with time but it is large for small time lags. The changes of MICA with the considered domain are similar to the ones of the other indices.

   – OIDRA: The index OIDRA is the most noise-safe index, having percentile differences $\Delta p$ smaller than 1 for both conditions (1) and (2). It correctly depends on the proximity of the objects, and the change of OIDRA is significant for large shifts of the extra object. OIDRA is also the index that is less sensitive to different resolutions. Specifically,

even after a change in resolution of 6 times, its values remain correlated at 0.9 with those of the original resolution. The autocorrelation of OIDRA is close to 1 for small shifts in time and it decreases rapidly, reaching less than 0.2 in 10 hours. The correlation of OIDRA with itself during a domain shift is nearly 1 for small shifts, gradually decreasing initially and then exhibiting a more rapid decline for larger shifts. The index OIDRA is the best index according to the conditions studied in this work, except for condition (4). It increases when an object is increasing only if the size of that object is

larger than the average mean size. Oppositely, if the size of that object is smaller than the average, OIDRA decreases. This behavior can be avoided by comparing OIDRA within bins of total convective area.

Among the indices considered in this work, none satisfies all the desired conditions. Besides, some of the studied conditions are only poorly satisfied by the indices. Thus, two main conclusions can be extracted from these results. First, the unsatisfied conditions and the different behavior that emerged from this study can explain the disagreement between different studies on

organization. Second, the indices here studied may not be enough to completely characterize organization, and a more complete metric could be built by simultaneously using more indices as suggested by Pscheidt et al. (2019) and performed by Janssens et al. (2021).





## 6 Outlook

The results of this assessment provide the first step to estimating uncertainties in quantifying convective organization. In the
470 following, we discuss possible extensions of this study.

This article focuses on a domain of $10°\text{x}10°$ because it is comparable to the domain sizes used in cloud-resolving model
studies (Tompkins, 2001; Bretherton et al., 2005; Muller and Held, 2012; Wing and Emanuel, 2014; Holloway and Woolnough,
2016). In recent years, several observational studies tried to quantify the convective organization on the entire tropics (Xu et al.,
2019; Bony et al., 2020; Jin et al., 2022; Bläckberg and Singh, 2022; Stubenrauch et al., 2023), hence a similar assessment
should be performed using the tropics as a domain. Such a domain is far from being a square, thus attention should be given
to the influence of domain shape and size. Furthermore, the tropical band is composed of regions with different degrees of
convective homogeneity, therefore, it may be interesting to study the influence of the spatial distribution of convection on
organization. Other effects to be studied include the smoothing of the data (Bony et al., 2020; Bläckberg and Singh, 2022), as
well as the variables and thresholds used to identify convection Stubenrauch et al. (2023), because these affect the shape and
480 the disposition of the objects.

## Appendix A: Definitions of the organization indices

The indices under study are briefly described below. The specific case of a sole aggregate in the domain is not considered. In
the following, the number of objects under study is indicated by $N$, the area of the i-th object is $A_i$ and the distance between
the centroids of the i-th and the j-th object is $d_{ij}$. The characteristic domain length is called $L$ and the total image size is $L^2$.

### A0.1 Organization index

The organization index ($I_{\text{org}}$ ) (Weger et al., 1992; Tompkins and Semie, 2017) is derived from the comparison of two dis-
tributions. Let $\hat{F}(d_{nn})$ be the cumulative distribution of the nearest neighbor distance $d_{nn}$ of the objects under study. The
cumulative distribution of the nearest neighbor distance of point-like objects randomly displaced in an unbounded domain is
490 a Weibull distribution $F(d_{nn}) = 1 - \exp(-\lambda \pi d_{nn}^2)$, where $\lambda$ is the mean number of objects per unit area. The organization
index is then defined by

$$I_{org} = \int_0^1 \hat{F}(F^{-1}(x))dx \tag{A1}$$

that is the area under the curve $(F(d_{nn}), \hat{F}(d_{nn}))$ between 0 and 1. The value of $I_{org}$ is between 0 and 1, and it is close to 0.5
for random distributions.



### A0.2 The index $L_{org}$

The index of organization $L_{org}$ (Biagioli and Tompkins, 2023) is computed by comparing the distribution of all-neighbor distances. The cumulative distribution of all-neighbor distances is represented by Ripley's function (Ripley, 1976, 1977, 1981)

$$K(r) = \mathbb{E}(N(b(\mathbf{x},r)\,\mathbf{x}) \tag{A2}$$

where $\mathbb{E}(N(b(\mathbf{x},r)\,\mathbf{x})$ is the expectation value of the number of objects in a dist $b(\mathbf{x},r)$ centered in $\mathbf{x}$ and with radius $r$, excluding $\{\mathbf{x}\}$. The Besag's L-function (Besag, 1977) is defined as $L(r) = \sqrt{K(r)/\pi}$, and its value for point-like objects randomly displace in an unbounded domain is $L(r) = r$. Let $\hat{L}(r)$ be the observed Besag's L-function, and $L^{th}(r) = r$ be the theoretical one. The index $L_{org}$ is

$$L_{org} = \frac{1}{r_{max}} \int\limits_0^{r_{max}} [\hat{L}(r) - L^{th}(r)]dr \tag{A3}$$

where $r_{max}$ is the integration limit, and it has to be equal to the greater possible distance between objects' pairs.

### A0.3 Simple Convective Aggregation Index

The Simple Convective Aggregation Index (SCAI) (Tobin et al., 2012) is based on the number of and the distances between objects. Let $D_0$ be the geometric averaged distance between each paired object's centroid $D_0 = \sqrt[\frac{N(N-1)}{2}]{\Pi_{i,j}d_{ij}}$. SCAI is defined by

$$\mathrm{SCAI} = k\frac{ND_0}{L^3} \tag{A4}$$

where $k$ is a constant. It was originally set to 2000, but it does not affect the result.

### A0.4 Modified Convective Aggregation Index

The Modified Convective Aggregation Index (MCAI) (Xu et al., 2019) is a modification of SCAI that also takes into account the areas of the objects under study. Similarly to SCAI, MCAI is defined by

$$\mathrm{MCAI} = k\frac{ND_2}{L^3} \tag{A5}$$

where $k$ is a constant, and $D_2$ is the arithmetical average of size corrected distance between objects

$$D_2 = \frac{2}{N(N-1)} \sum_{i=1}^N \sum_{j=i+1}^N \max(0, d_{ij} - \sqrt{A_i/\pi} - \sqrt{A_j/\pi}) \tag{A6}$$



### A0.5 Convective Organization Potential

The Convective Organization Potential (COP) (White et al., 2018) is built on the concept of an interaction potential that tries
to reproduce the gravitational force among cloud systems in a bidimensional space. The definition of COP is given by

$$\text{COP} = \frac{2}{N(N-1)} \sum_{i=1}^{N} \sum_{j=i+1}^{N} \frac{\sqrt{A_i/\pi} + \sqrt{A_j/\pi}}{d_{ij}} \tag{A7}$$

which is the mean over all the possible pairs of the interaction potential.

### A0.6 Area-based Convective Organization Potential

The Area-based Convective Organization Potential (ABCOP) (Jin et al., 2022) is a modification of COP. It is defined by

$$\text{ABCOP} = \frac{1}{2L} \sum_{i=1}^{N} \max_{j \neq i} \left( \frac{A_i + A_j}{d_2(i,j)} \right) \tag{A8}$$

where $d_2(i,j) = \max(1, d_{i,j} - \sqrt{A_i/\pi} - \sqrt{A_j/\pi})$ is an estimate of the distance between the edges of the i-th and the j-th
object.

### A0.7 Radar Organization Metric

The Radar Organization Metric (ROME) (Retsch et al., 2020) was originally defined to measure the strength of organization
within a radar scene. Let $\tilde{d}_{ij}$ be the smallest distance between the edges of the i-th and the j-th object in the domain. ROME is
defined by

$$\text{ROME} = \frac{2}{N(N-1)} \sum_{i=1}^{N} \sum_{j=i+1}^{N} \left[ A_{ij}^{(max)} + A_{ij}^{(min)} \cdot \min\left( 1, \frac{A_{ij}^{(min)}}{\tilde{d}_{ij}^2} \right) \right] \tag{A9}$$

where $A_{ij}^{(max)} = \max(A_i, A_j)$ and $A_{ij}^{(min)} = \min(A_i, A_j)$. The value of ROME is always in between $\bar{A} < ROME < 2\bar{A}$,
where $\bar{A}$ is the object mean size $\sum_i A_i / N$.

### A0.8 Morphological Index of Convective Aggregation

The Morphological Index of Convective Aggregation (MICA) (Kadoya and Masunaga, 2018) is the only index that takes into
account both the cloud and the clear sky coverage. Let $A_{cld}$ be the area of the smaller rectangle that encloses all the objects
under study in the domain. MICA is defined by

$$\text{MICA} = \frac{\sum_i A_i}{A_{cld}} \cdot \frac{|L^2 - A_{cld}|}{L^2} \tag{A10}$$

which is the multiplication of two terms: the first term quantifies the density of objects within a confined area, and the second
term quantifies the amount of clear sky in the studied domain.



## A0.9 Organization Index based on Distance and Relative Area

A new index of organization called Organization Index based on Distance and Relative Area (OIDRA) is defined and given as
an example of a well-behaving index. Let $\tilde{d}_{ij}$ be the smallest distance between the edges of the i-th and the j-th object in the
domain, and let $A_T = \sum_i A_i$. OIDRA is defined by

$$
\text{OIDRA} = \frac{1}{A_T^2} \sum_{i=1}^{N} A_i^2 + \frac{2}{A_T^2} \sum_{i=1}^{N} \sum_{j=i+1}^{N} A_i A_j \left( 1 - \sqrt{\frac{\tilde{d}_{ij}}{l}} \right) \tag{A11}
$$

where $l$ is the length scale at which we desire to study organization and it is set to $L/\sqrt{2}$ in this study. The first part takes
into account the different object sizes, while the second part depends also on the distances between them. OIDRA introduces
two new concepts that have never been used before in other oganization indices. Firstly, OIDRA does not depend explicitly on
the size of the objects, but only on their relative fraction. Secondly, the parameter $l$ can be set to different values in order to
calculate organization at different scales. When two objects are in close proximity ($\tilde{d}_{ij} << l$), their contribution to the OIDRA
is similar to what it would be if the objects were merged. When two objects are very far ($\tilde{d}_{ij} >> l$), their contribution becomes
negative. This index is always smaller than 1, and it is positive if $l > L/\sqrt{2}$. In our case ($l = L/\sqrt{2}$), OIDRA varies between 0
and 1.

*Code and data availability.* The TOOCAN dataset is available at the official website https://toocan.ipsl.fr/. The code developed to perform
this study is available at https://zenodo.org/record/8287752. The random dataset can be generated with the available code, and the entire
analysis can be run using the generated dataset. All the plots that the code is producing are already available at
https://web.lmd.jussieu.fr/ gmandorli/Assessment_of_the_object-based_indices_of_convective_organization.

*Author contributions.* GM developed the concept, undertook the analysis, and prepared the figures. CJS contributed to the discussion of the
analysis. Both CJS and GM contributed to the manuscript writing.

*Competing interests.* The authors declare no competing interests.

*Acknowledgements.* This work was supported by the Centre National de la Recherche Scientifique (CNRS), the Centre National d'Etudes
Spatiales (CNES), and the TTL-Xing ANR-17-CE01-0015 project. The authors express their gratitude to Thomas Fiolleau for his invaluable
assistance in comprehending and utilizing the TOOCAN dataset. The authors extend their appreciation to Giovanni Biagioli for engaging in
constructive discussions regarding the behavior of the organization indices.





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
