# Peer review of "Assessment of object-based indices to identify convective organization"

_EGUsphere, 2023_

## Referee Comment (RC1)

Manuscript: egusphere-2023-1985

**Assessment of object-based indices to identify convective organization**

The manuscript is well-structured, providing a comprehensive review of past convective indices and offering a quantitative comparative analysis of nine object-based indices based on a set of criteria. The authors introduce a promising index that they suggest is a 'well-behaving index' according to the defined criteria. Nevertheless, this reviewer has a few major concerns that must be addressed for publication. I encourage the authors to view these comments as an opportunity for improvement.

One significant concern is the absence of a proper review and utilization of other existing trackers for convective system identification. Given the manuscript's focus on comparing nine object-based indices, it is both reasonable and, to some extent, ethical to incorporate at least two (if not more) MCS tracking algorithms. This addition would enhance the robustness of the results. Furthermore, a recent MCS-tracking intercomparison study by Prein et al. 2023 (https://essopenarchive.org/doi/full/10.22541/essoar.169841723.36785590) has documented relevant differences in results related to MCS characteristics and statistics across different tracking algorithms.

In addition to the generated dataset, it is suggested that the authors incorporate at least one more tracker in the analysis to ensure multiple algorithms contribute to convective system identification data. One suitable option is TAMS (Núñez Ocasio et al. 2020a; Núñez Ocasio et al. 2020b; https://tams.readthedocs.io/en/latest/), an objective MCS tracking algorithm. TAMS is open-source, publicly available, and Python-based, making it a viable candidate for comparison with TOOCAN.

Both TOOCAN and TAMS share underlying similarities in identification and tracking, yet they differ sufficiently for a comprehensive comparative analysis. Similar to TOOCAN, TAMS utilizes Tb, allowing authors to download satellite Tb for the warm pool region domain in case TOOCAN systems cannot be separated from the Tb data. Additionally, like TOOCAN, TAMS allows saving the mask for the identified convective objects.

For further reference, authors are encouraged to refer to Prein et al. 2023 for information on other trackers that could be considered, such as MOAAP and PyFLEXTRKR. It is advised to provide a proper review of MCS trackers as convective system identification algorithms, including MOAAP, PyFLEXTRKR, and TAMS (in addition to TOOCAN), which are all current open-source MCS trackers available.

Because of the intrinsic relationship between deep convective organization and how an MCS is defined or identified (the first step of a tracking algorithm), this manuscript would benefit from the inclusion of a discussion regarding how the new index is sensitive to the MCS tracking algorithm being used and vice versa. How does the sensitivity of the new 'well-behaving index' compare to the sensitivity of other indices to multiple MCS trackers?

Another concern is that the manuscript has some technical English errors that should be addressed. Mainly, these are grammatical aspects that can hinder the interpretation of the discussion. This reviewer has pointed out some of them below in the introduction but there are

more throughout the paper. Authors are encouraged to thoroughly revise the manuscript with the guidance of an internal technical writing reviewer before resubmitting.

**Specific comments:**

Line 16: Instead of "is the main", "is primarily"

Line 16: Thus,

Line 17: "we talk about", is too colloquial, please rephrase.

Line 18: "On *the* climate"

The introduction would benefit from a review of what is convection and convective organization. Although it does not have a rigorous definition, certainly, past papers must have addressed convective organization that is relevant to include here to introduce such indices.

Lines 59-65: This methodology is not clear. Why tune the generated dataset to TOOCAN? Doesn't seem to be a fair comparison then. Please address.

Lines 110-115: This is confusing, are the authors referring to the numbers in Table 1? They are all way above or way below 0.5. As the author pointed out, it is incoherent. Is there a clearer way to represent these numbers?

For Condition 7: What happens if a study has a continuous domain? Like uninterrupted global datasets? Will the results change?

The quantitative statistical analysis provided by distributions is well structured and complete.

The final summary for each index is very informative and provides a succinct summary of the results. The summary reveals relevant information regarding these emerging indices and the relevance in choosing the right indices for the right scientific question.

Can the authors provide some additional discussion on which of the indices compared the most with OIDRA?

---

## Referee Comment (RC2)

**Review**
The authors investigate the robustness of 9 single number indices representing the degree of organization of convection (DOC) in a 10x10 degree domain. The convective regions are identified from thresholds on brightness temperature from geostationary satellite data (TOOCAN). The robustness of an organization metric is assessed from a sensitivity test on three categories of criteria; sensitivity to noise, sensitivity to changes in position and size of contiguous convective regions (8-connected convective gridboxes), and sensitivity to specifics of the dataset (mainly temporal and spatial resolution of sampling). The study addresses central questions surrounding the quality and versatility of different organization metrics and presents a new metric with high degree of robustness from these criteria on the specified domain. Furthermore, the study serves as a summary of current methods in assessing DOC and a great foundation for further improvements of organization metrics.

The sections of the paper are well structured, with informative methodology, illustrative examples, and concise explanation of the results.

The reviewer has a few minor concerns / questions that need to be addressed before approval of publication:

**Conceptual**
*1. I_org assessment*
It is interesting that the I_org metric is so sensitive in all categories of robustness criteria. I wonder to what extent the sensitivity relates to the number of objects considered. Is I_org robust when considering a large number of objects? How large fraction of the scenes considered in this study have less than 35 objects (where the I_org metric is no longer reliable)?

The I_org metric has been identified as unreliable for a small number of convective centroids (<20) before:
https://agupubs.onlinelibrary.wiley.com/doi/full/10.1029/2019GL086927
Not in a systematic way as in the present study, but perhaps it should still be mentioned.

In several studies, I_org is used to assess aggregation from local minima in brightness temperature, which includes multiple convective centroids (convective cores) in a large convective object. With this approach the area of convective objects is implicitly included (as a large convective object introduce several closely connected convective cores), and the issue with a small number of convective centroids is addressed. Perhaps assessing I_org from the convective cores approach can be insightful to better understand the utility of this metric. Otherwise, perhaps just presenting the alternative approach and clarifying that the statement in the present study relates to the method applied to convective objects.
In this study:
https://agupubs.onlinelibrary.wiley.com/doi/full/10.1029/2019AV000155
the authors mention that a similar result is obtain from using the object-based approach and the convective cores approach. Albeit, in that study the domain is the whole tropics, so the number of convective centroids is likely sufficiently large regardless.

*2. ROME assessment*
From working with the ROME metric to assess the tropical domain with DOC, I know that the metric is highly correlated with mean area of the domain. It was interesting to see that the impact of changes to the proximity of convective regions was so small. I suspect changes to the proximity of convective objects has a greater relative effect on the metric if the scenes are sub-sampled to scenes with similar energetic constraints (similar mean convective area, similar vertical velocity, similar mean precipitation etc.). Perhaps the distance component of ROME is also more significant when very large distances are considered (where the squared edge distance is a larger number of multiples of the smaller pair object). Further, considering that the proximity scaling applies to every object pair, a larger number of objects all moving together may also highlight the proximity scaling. I reserve the possibility, that the metric simply is unable to factor in the proximity of objects as it occurs in realistic settings, past the change in proximity which results in joining two objects. However, it would be interesting to test some of these considerations to highlight the limitations / utility of this metric.

*3. ABCOP assessment*
In the conclusion, it sounds like it is recommended to use this metric. While the metric captures changes in proximity, and is robust in most criteria, from what I understand, the metric does not correctly capture fundamental changes in aggregation; adding a random single convective gridbox increases aggregation, and merging objects decreases aggregation, which are the opposite signs of change from the conceptual interpretation. Perhaps it can be highlighted that these features make the metric unsound in this regard.

**Technical**
When testing condition 4 (changing the size of one object) - from the schematic in figure 6, it appears that the edge of the test object effectively move closer to the other objects when the area of the test object increases. Consequently, there will be a proximity component affecting metrics that depend on the edge distance between objects. To avoid a proximity influence when testing condition 4 on ABCOP and ROME, the test object could be uniformly extended Eastward in these cases.

In the methodology section it was mentioned that scenes with one or no objects were removed. What fraction of scenes contain only one object, and are they significant for describing degree of organization? Further, are they small objects or very large objects spanning most of the domain. Is it important for a metric to be able to handle a singular large object for the 10x10 degree domain?

In the introduction in line 27 there is a statement: 'Such studies have been so far performed only for example cases'. It could be nice with a reference to this. In the ROME metric paper, for example, there is an evaluation of different organization metrics for example cases, so maybe that paper can be referenced here:

**Language**
There are some instances of grammatical errors, and some typos. Here are a few I noticed:
[line: 'instance' - suggestion]
Line 16: 'Convection is the main responsible' - mainly responsible
Line 26: 'organization has not a rigorous' - organization does not currently have a rigorous
Line: 33: 'Moreover, it briefly recalls the convective organization indices' - Sect. 2.2 (it was hard to interpret what 'it' was referring in this line)
Line 212: 'more peaked' - exhibit a narrower peak around zero
Line 435-436: 'ROME is the index that depends less on' - is the least sensitive to
Line 454: 'OIDRA is also the index that is less sensitive' - the least sensitive
Line 459: 'It increases when an object is increasing' - 'when an object is increasing in size' (I think there are a couple more instances similar to this)
Line: 460: 'larger than the average mean size' - repetition, so keep one
Line 200: 'Move from 0.91 and 0.78' - Move from 0.91 to 0.78'
Line 303: figure 8 description: 'with a 3 times worst resolution' - coarse-grained by a factor of 3
Line 435: 'ROME depends very much more on the size' - is highly dependent on the size

---

## Author Comment (AC1)

**Responses to comments of "Assessment of object-based indices to identify convective organization" egusphere-2023-1985 to GMD**

We express our gratitude to the referee for providing constructive criticisms and valuable comments which have been very helpful in improving the quality of this manuscript. We have made the point-by-point response to the comments and revised the manuscript accordingly.

We hope that the revised version can obtain approval and meet the journal's requirements. In this document, the referee's comments are presented alongside our responses (in blue) and the textual modifications (in red). Both authors have thoroughly reviewed the revised manuscript and unanimously agreed to its submission in this improved form.

**Request**

One significant concern is the absence of a proper review and utilization of other existing trackers for convective system identification. Given the manuscript's focus on comparing nine object-based indices, it is both reasonable and, to some extent, ethical to incorporate at least two (if not more) MCS tracking algorithms. This addition would enhance the robustness of the results. Furthermore, a recent MCS-tracking intercomparison study by Prein et al. 2023 (https://essopenarchive.org/doi/full/10.22541/essoar.169841723.36785590) has documented relevant differences in results related to MCS characteristics and statistics across different tracking algorithms.

**Answer**

We thank the reviewer for this comment, which highlighted that the data description is very misleading in the manuscript.

In the current analysis, we do not make use of the TOOCAN tracking algorithm. We have only used the brightness temperature measurement calibrated by Fiolleau et al. (2020) which also led to the construction of the TOOCAN systems.

We have modified the manuscript to clarify this point, and we decided not to use the word TOOCAN to refer to the dataset.

**Changes in Manuscript (line 37)**

**2.1 Datsets of convective objects**

The statistical comparison between indices needs a dataset of horizontal binary fields that mimic deep convective clouds for which it is possible to compute the convective organization indices.

Since the goal is not to study physical processes but the behavior of the indices, any dataset can be used. However, in order to well represent the typical size, occurrence, and disposition of deep convection in the tropics, we have chosen a real satellite dataset with a good spatial and temporal resolution.

Fiolleau et al. (2020) provide such a dataset with calibrated infra-red (IR) brightness temperatures ($T_B$) by combining different geostationary satellites to span the entire band from 40N to 40S. The spatial resolution is 0.04°, and the temporal frequency is 30 minutes. For this study, we reconstruct convective objects from cold brightness temperatures with a cold core ($T_B < 190$ K) surrounded by $T_B < 235$ K, by grouping all 8-connected grid boxes. Holes in each object are filled to avoid degenerate dispositions. This procedure is implemented with the Python framework developed by van der Walt et al. (2014). We selected the oceanic tropical Warm Pool region expanding over 0°N-9.6°N and 140.4°E-150°E. The original resolution is downscaled to 0.08° to analyze images with a size of 120x120 grid boxes. Then, images with less than two objects are rejected. Finally, a total of 76462 images in the period 2012-2016 is considered for this study.

**Request**

In addition to the generated dataset, it is suggested that the authors incorporate at least one more tracker in the analysis to ensure multiple algorithms contribute to convective system identification data. One suitable option is TAMS (Núñez Ocasio et al. 2020a; Núñez Ocasio et al. 2020b; https://tams.readthedocs.io/en/latest/), an objective MCS tracking algorithm. TAMS is open-source, publicly available, and Python-based, making it a viable candidate for comparison with TOOCAN.

Both TOOCAN and TAMS share underlying similarities in identification and tracking, yet they differ sufficiently for a comprehensive comparative analysis. Similar to TOOCAN, TAMS utilizes Tb, allowing authors to download satellite Tb for the warm pool region domain in case TOOCAN systems cannot be separated from the Tb data. Additionally, like TOOCAN, TAMS allows saving the mask for the identified convective objects.

For further reference, authors are encouraged to refer to Prein et al. 2023 for information on other trackers that could be considered, such as MOAAP and PyFLEXTRKR. It is advised to provide a proper review of MCS trackers as convective system identification algorithms, including MOAAP, PyFLEXTRKR, and TAMS (in addition to TOOCAN), which are all current open-source MCS trackers available.

Because of the intrinsic relationship between deep convective organization and how an MCS is defined or identified (the first step of a tracking algorithm), this manuscript would benefit from the inclusion of a discussion regarding how the new index is sensitive to the MCS tracking algorithm being used and vice versa. How does the sensitivity of the new 'well-behaving index' compare to the sensitivity of other indices to multiple MCS trackers?

**Answer**

We appreciate the referee for bringing up this important topic, which holds particular significance for those delving into the study of convective organization applied to MCS.

Indeed, different tracking algorithms can identify different convective systems, thus they can produce different values of convective organization indices. A deep study of such a sensitivity is crucial and it should be a focal point for future investigations.

A first international workshop on Cloud Tracking was already held in April 2023 (hosted in Oxford by Philip Stier), and a Joint NASA (AOS) – INCUS – GEWEX Convection Tracking workshop is planned for April 2024 in New York.

Our current work is independent and complementary to the assessment of the tracking algorithms. The primary objective of this study is to analyze the consistency of the convective organization indices. The identification of well-behaving indices equates to discerning which indices consistently reflect convective organization. It is important to stress that only well-behaving indices offer a reliable measurement of organization, thus they are the only ones that can be used in climate analyses, regardless of the MCS tracker employed. However, we have added a small paragraph in the introduction to make this clearer.

The sensitivity of the indices to different tracking algorithms is instead something very different because it does not assess the consistency of the indices. This type of test depends also on the algorithms, and it should be performed only after having a complete well-behaving index (one that fulfills all seven conditions).

**Changes in Manuscript (line 31)**

The convective objects have been identified by images of continuous areas of cold infrared brightness temperature measurements. This assessment is complimentary and independent of the assessment of convection tracking methods (e.g. Prein et al., 2023), which have been developed to identify the convective objects.

**Request**

The introduction would benefit from a review of what is convection and convective organization. Although it does not have a rigorous definition, certainly, past papers must have addressed convective organization that is relevant to include here to introduce such indices.

**Answer**

The reviewer is certainly right. We have expanded the introduction of the manuscript by adding a small review of convection and convective organization.

**Changes in Manuscript (line 16-23)**

Atmospheric convection is a fundamental process characterized by the vertical movement of air masses within the Earth's atmosphere. As the sun heats the Earth's surface, warm air rises, transporting heat and moisture through the atmosphere. This upward motion triggers the formation of clouds and weather phenomena, playing a crucial role in shaping our planet's weather and climate. In Radiative-convective equilibrium simulation, convection shows a tendency to cluster horizontally as time passes. This behavior was firstly pointed out by Held et al. (1993), and then it was confirmed in several other studies (Tompkins, 2001; Bretherton et al., 2005; Wing

and Emanuel, 2014). Because of this feature, clustered convection is referred to as aggregated or organized convection, or convective organization. In recent years, because of the great importance of convection on climate, many studies have been focusing their attention on convective organization. Either looking for an explanation of such a phenomenon with simulation (Wing and Emanuel, 2014; Tompkins and Semie, 2017; Cronin and Wing, 2017; Muller and Romps, 2018; Muller et al., 2022) or trying to measure convective organization in observations and relate it to known quantities (Wing et al., 2017, 2020; Bony et al., 2020; Bläckberg and Singh, 2022; Stubenrauch et al., 2023). Both types of analysis need a method to quantify convective organization. However, quantifying the degree of convective organization is challenging. There is still no consensus on the best method to use and various methods have been proposed in recent years, reviewed by Biagioli and Tompkins (2023).

**Question**

Lines 59-65: This methodology is not clear. Why tune the generated dataset to TOOCAN? Doesn't seem to be a fair comparison then. Please address.

**Answer**

The reviewer has highlighted a crucial point that has been neglected in the manuscript.

The behavior of certain indices can be dependent on the number of objects ($N$), for example as shown in Fig. 5. Consequently, they can depend on the distribution of $N$.

When comparing different datasets, several differences may emerge. Some can be caused by the inherent nature of the datasets, including the shape and spatial distribution of objects, while others arise just from different distributions of $N$.

In this manuscript, our primary focus was on addressing the former, as they bear a more direct relevance to the intrinsic concept of convective organization. Differences attributed to $N$ were not considered within the scope of this article. Therefore, in order to get rid of any difference due to $N$ we have simulated a dataset in such a way that it reproduces the same distribution of the convective object dataset constructed from cold $T_B$.

Similarly, the object size can affect the indices behavior as well, therefore we simulated the dataset to reproduce also the distribution of object size.

This is also one of the reasons why comparing different tracking algorithms is complex and beyond the scope. Different tracking algorithms may strongly affect the indices behavior via their $N$, thus, extracting the differences coming from the algorithms itself is challenging.

**Changes in Manuscript (line 58-63)**

The following analysis aims to study the behavior of the organization indices, and the results shall not be dependent on the dataset used. The robustness of this analysis against the dataset can be proved by comparing the results obtained using different datasets. When comparing datasets, several differences may emerge. Some can be caused by the inherent nature of the datasets, including the shape and spatial distribution of objects, while others arise just from different distributions of objects number and sizes. The primary focus of this work is addressing the

former, as they bear a more direct relevance to the intrinsic concept of convective organization. To prove the reliability of the results here presented, we have simulated a dataset to compare with the convective object dataset obtained from cold brightness temperatures. Therefore, we have built images of randomly placed circular objects of different sizes. We used a Monte Carlo simulation technique which follows distributions of object sizes and number of objects with the same shape as the ones of the convective object dataset from cold $T_B$. Examples of images generated with this method are given in the supplement material. Despite the large differences in shape and spatial distribution of the objects in the two datasets, the final results are similar, meaning that they don't depend on the nature of the objects. The results obtained with the brightness temperature database are shown in the following, while the ones obtained with the newly simulated dataset are shown in the supplement material.

**Question**

Lines 110-115: This is confusing, are the authors referring to the numbers in Table 1? They are all way above or way below 0.5. As the author pointed out, it is incoherent. Is there a clearer way to represent these numbers?

**Answer**

Lines 110-115 refer to Table 1.

We agree with the referee that such a big table looks unclear. To facilitate the reading, we have modified the Table as follows:
- Numbers exceeding 0.5 correlation are presented in bold for emphasis,
- Correlations located below the diagonal have been omitted,
- A demarcating line has been added to distinguish the indices from the other variables.

Furthermore, we have updated the table caption to align with these improvements.

**Changes in Manuscript:**

Table 1: Correlation coefficients, multiplied by 100, of the indices with each other and with number, total area, and mean size of convective objects. Bold numbers highlight correlations with coefficients larger than 0.5.

| | $I_{org}$ | $L_{org}$ | COP | ABCOP | ROME | SCAI | MCAI | MICA | OIDRA | number | total area | mean size |
|---|---|---|---|---|---|---|---|---|---|---|---|---|
| $I_{org}$ | **100** | **74** | 38 | -15 | -25 | 35 | 31 | 43 | 10 | -23 | -33 | -26 |
| $L_{org}$ | | **100** | 47 | -16 | -16 | 41 | 40 | **56** | 22 | -26 | -27 | -16 |
| COP | | | **100** | -1 | 39 | 47 | 50 | **72** | 48 | -43 | 1 | 39 |
| ABCOP | | | | **100** | 47 | -34 | -31 | -13 | 39 | 33 | **81** | 46 |
| ROME | | | | | **100** | 5 | 10 | 14 | **52** | -10 | **68** | 100 |
| SCAI | | | | | | **100** | **99** | 49 | 31 | **-96** | -48 | 5 |
| MCAI | | | | | | | **100** | **51** | 34 | **-96** | -43 | 10 |
| MICA | | | | | | | | **100** | 49 | -43 | -19 | 13 |
| OIDRA | | | | | | | | | **100** | -29 | 39 | **51** |

**Question**

For Condition 7: What happens if a study has a continuous domain? Like uninterrupted global datasets? Will the results change?

**Answer**

Condition 7 applies to open domains (i.e. with defined borders), which are what most of the studies of convective organization are targeting. Some studies instead consider uninterrupted domains, like CRM with double periodic conditions or global datasets. For such domains, perturbation 7 is ineffective and the indices should not change when the domain is shifted. Therefore, condition 7 is satisfied automatically. As a consequence, indices that do not satisfy condition 7 cannot be used on open domains but they still can be used on continuous domains.

**Request**

Can the authors provide some additional discussion on which of the indices compared the most with OIDRA?

**Answer**

We have added a discussion of OIDRA in comparison with the other indices in the appendix, just after explaining of the properties of OIDRA's definition.

**Changes in Manuscript:**

Because of its specific formulation, OIDRA is different from all the other indices. The main reason can be attributed to its dependence on the object sizes, which are squared. This feature makes OIDRA very sensible to object sizes, which makes it similar to ROME. ROME and OIDRA exhibit similar behaviors for conditions 1, 2, and 5, where the object size plays a crucial role. Moreover, ROME and OIDRA correlate higher than 0.5. Nevertheless, OIDRA's response to object proximity aligns more closely with $L_{org}$ than with other indices.

Sincerely,

Giulio Mandorli,
Claudia Stubenrauch

---

## Author Comment (AC2)

February 13, 2024

**Responses to comments of "Assessment of object-based indices to identify convective organization" egusphere-2023-1985 to GMD**

We deeply thank the referee for the very insightful and helpful comments which show that he did understand very well our work. The suggestions offered have resulted in revisions to the manuscript, which enhances its overall quality.

In the following, we have made a point-by-point response to the comments and revised the manuscript accordingly. The referee's comments are presented alongside our responses (in blue) and the textual modifications (in red). Both authors have thoroughly reviewed the revised manuscript and unanimously agreed to its submission in this improved form.

**1 Conceptual**

**Comment**

*1. I_org assessment*
It is interesting that the I_org metric is so sensitive in all categories of robustness criteria. I wonder to what extent the sensitivity relates to the number of objects considered. Is I_org robust when considering a large number of objects? How large fraction of the scenes considered in this study have less than 35 objects (where the I_org metric is no longer reliable)?

The I_org metric has been identified as unreliable for a small number of convective centroids (<20) before:
https://agupubs.onlinelibrary.wiley.com/doi/full/10.1029/2019GL086927
Not in a systematic way as in the present study, but perhaps it should still be mentioned.

**Answer**

We thank the referee for suggesting the work of Semie and Bony (2020) that has been cited in the manuscript.

In some tests, the sensitivity strongly depends on the number of objects, for example, conditions 1 and 2. For this reason, the sensitivity has been shown as a function of the number of objects in Fig.5. On the other hand, the results shown in Fig.4 and Table 1 include all events therefore they are averaging events where $N < 35$ and where $N > 35$.

In some other tests, the sensitivity does not depend very much on the number of objects (see later), and for those tests $I_{org}$ can be robust also where $N < 35$. Therefore, we did not show any results as a function of the number of objects similar to Fig.5.

The fraction of scene as a function of the number of object can be roughly seen in Fig.1. Scenes where $N < 35$ ($N > 35$) are 93% (7%) of the total.

**Changes in Manuscript (line 231)**

The index $I_{org}$ is very sensible to noise at a low number of objects, and it becomes noise-safe when more than 35 objects are present. This result is in agreement with Semie and Bony (2020) who raised a similar statement of robustness for $I_{org}$.

**Comment**

In several studies, I_org is used to assess aggregation from local minima in brightness temperature, which includes multiple convective centroids (convective cores) in a large convective object. With this approach the area of convective objects is implicitly included (as a large convective object introduce several closely connected convective cores), and the issue with a small number of convective centroids is addressed. Perhaps assessing I_org from the convective cores approach can be insightful to better understand the utility of this metric. Otherwise, perhaps just presenting the alternative approach and clarifying that the statement in the present study relates to the method applied to convective objects.
In this study:
https://agupubs.onlinelibrary.wiley.com/doi/full/10.1029/2019AV000155
the authors mention that a similar result is obtain from using the object-based approach and the convective cores approach. Albeit, in that study the domain is the whole tropics, so the number of convective centroids is likely sufficiently large regardless.

**Answer**

We thank the referee for raising such an important point that was not mentioned in the manuscript.

In this work, we have used convective object reconstruction. The conditions 1-7 are designed to be satisfied for objects with a real size (e.g. condition 2), and all the presented results relate to this approach. This is the only approach that can be used to directly compare so many indices of organization. On the other hand, $I_{org}$ and $L_{org}$ can also be applied to local minima, more related to convective cores. However, such a method supposes point-like objects, and therefore it precludes the possibility to compute all the other indices.

Following the referee's remark, $I_{org}$ and $L_{org}$ can be applied to both local minima and convective objects. In this case, they can have different behaviors. Such a clarification has been added in the manuscript in the conclusions when all the results are presented for each index.

Lastly, the referee's question is part of a wider question, that is how to define the objects under study. There are many possibilities: mesoscale convective systems defined by brightness temperature, deep convective cores defined by strong precipitation, convective cores defined by local minima in brightness temperature, and many others. This question would lead to a new assessment, which focuses only on object definition.

**Changes in Manuscript (line 396)**

The above-mentioned behaviors of $I_{org}$ are obtained using convective object reconstruction. However, since $I_{org}$ does not consider the size of the objects, some studies (e.g. Semie and Bony, 2020; Bony et al., 2020) applied it to local minima in brightness temperature which may be seen as a proxy for the convective core positions. Such an approach may modify the behavior of this index. Identical considerations apply to $L_{org}$.

**Changes in Manuscript (line 401)**

Equivalently to $I_{org}$, the present results hold only when convective objects are reconstructed.

**Comment**

*2. ROME assessment*
From working with the ROME metric to assess the tropical domain with DOC, I know that the metric is highly correlated with mean area of the domain. It was interesting to see that the impact of changes to the proximity of convective regions was so small. I suspect changes to the proximity of convective objects has a greater relative effect on the metric if the scenes are sub-sampled to scenes with similar energetic constraints (similar mean convective area, similar vertical velocity, similar mean precipitation etc.). Perhaps the distance component of ROME is also more significant when very large distances are considered (where the squared edge distance is a larger number of multiples of the smaller pair object). Further, considering that the proximity scaling applies to every object pair, a larger number of objects all moving together may also highlight the proximity scaling. I reserve the possibility, that the metric simply is unable to factor in the proximity of objects as it occurs in realistic settings, past the change in proximity which results in joining two objects. However, it would be interesting to test some of these considerations to highlight the limitations / utility of this metric.

**Answer**

Indeed it would be interesting to subsample the events and test the indices behaviors for each sample. We did not fully compute such a study, however, we checked a few cases and we report here a few examples. Fig.R1 displays the sensitivity test of condition 3 on four different subsamples of the dataset.

The upper panels show the proximity test for two bins of the number of objects. Despite the large difference in the number of objects, the results are similar. The lower panels show the proximity test for two bins of the total area of convection ($\sum_i A_i$). The bins of area of convection are about 4% and 40% of the domain under consideration. Of course, the results depend on the number of objects and the total area of convection, however, the dependency is small if compared with conditions 1 and 2.

**Comment**

*3. ABCOP assessment*
In the conclusion, it sounds like it is recommended to use this metric. While the metric captures

[Figure]

Figure R1: Following the referee's comments, the figure presents the sensitivity test to proximity (condition 3 of the manuscript) for 4 different subsamples of the data. The subsamples are: events with a number of objects within 10 and 15 (top left), events with a number of objects within 30 and 50 (top right), events with a total area of convective objects within 0.03 and 0.05 $Mm^2$ (bottom left), events with a total area of convective objects within 0.3 and 0.5 $Mm^2$ (bottom right).

changes in proximity, and is robust in most criteria, from what I understand, the metric does not correctly capture fundamental changes in aggregation; adding a random single convective gridbox increases aggregation, and merging objects decrease aggregation, which are the opposite signs of change from the conceptual interpretation. Perhaps it can be highlighted that these features make the metric unsound in this regard.

**Answer**

The referee does understand well the results, and he is right. ABCOP is well behaving in most criteria, but the two features he mentioned make it a bad candidate to spot convective organization.

We changed the conclusion to highlight more of those behaviors.

- ABCOP: The index ABCOP has a large correlation with the total objects' area $\sum_i A_i$. Therefore, it mostly reflects the total objects' area. This relationship uniquely comes from the larger objects in the images, while the smaller objects do not play a significant role in the value of ABCOP. This behavior comes from the $\max_{j \neq i}()$ function in the ABCOP definition (equation A8), which gives great importance to large objects. As a consequence, ABCOP is not very sensitive to noise. However, it strongly depends on the number of objects because it is defined as a sum over each object instead of as a mean like COP. This characteristic is visible in two of the studied conditions and it negatively influences ABCOP response to noise. First, when one convective grid box is added randomly in the domain, ABCOP incorrectly increases instead of decreasing because the number of objects is increasing. Second, when two close objects are merged, ABCOP incorrectly decreases instead of increasing because the number of objects decreases. In addition, ABCOP follows the behavior the number of objects even under a change of horizontal resolution. ABCOP proves to be robust under changes in resolution, shifts in time, and shifts of the considered domain. Last but not least, the index ABCOP increases with the proximity of the objects but slower than the majority of indices.

**2  Technical**

**Comment**

When testing condition 4 (changing the size of one object) - from the schematic in figure 6, it appears that the edge of the test object effectively move closer to the other objects when the area of the test object increases. Consequently, there will be a proximity component affecting metrics that depend on the edge distance between objects. To avoid a proximity influence when testing condition 4 on ABCOP and ROME, the test object could be uniformly extended Eastward in these cases.

**Answer**

The referee raised a good point. The perturbation of condition 4 modifies also the distances between objects for 4 indices: ABCOP, MCAI, ROME, and OIDRA. ABCOP and MCAI are affected because they use the effective distance $d_{eff} = d(i,j) - r_i - r_j$, where $r_i$ is the equivalent radii of the object. ROME and OIDRA directly use the distances between edges via the python package "shapely". For these four indices, the influence of proximity should be removed as suggested by the referee. However, if we implement such a correction, the results of the original test on $I_{org}$, $L_{org}$, COP, MICA, and SCAI cannot be compared with the results of the new test on ABCOP, MCAI, ROME, and OIDRA. Therefore, we decided to perform one single test on all the indices.

[Figure]

Figure R2: Sketch of the perturbation suggested by the referee (left) and the associated $< \Delta p >$ for all the indices (right).

To provide a complete answer to the referee, we have run the test the referee has suggested, and the results are shown in Fig.R2. For each index, the result can be compared with condition 4 shown in Fig.6 and Fig.7 of the manuscript.

The sensitivity to this new perturbation is similar to the one of condition 4. The main difference is that $I_{org}$, $L_{org}$, MICA, and SCAI have negative trends because are affected by the proximity component.

**Comment**

In the methodology section, it was mentioned that scenes with one or no objects were removed. What fraction of scenes contain only one object, and are they significant for describing the degree of organization? Further, are they small objects or very large objects spanning most of the domain.

**Answer**

Only 4.1% of the scenes contain no objects. Only 2.9% of scenes contain one object. Therefore we did not consider these 7% of the scenes.

The scenes with a small number of objects usually have very small objects. It is partially shown in Fig.1 of the manuscript, where the total coverage of convective objects is plotted as a function of the number.

**Changes in Manuscript (line 56)**

Then, images with less than two objects are rejected (4.1% and 2.9% of the events with no object and one object, respectively).

**Comment**

Is it important for a metric to be able to handle a singular large object for the 10x10 degree domain?

**Answer**

The referee asked a very meaningful question that does not have a well-defined answer. To date, there is no consensus on how a metric should behave when one singular large object spans the domain, and the authors do not have better insights on the topic than other experts. Such a question should be discussed in the scientific community.

**Comment**

In the introduction in line 27 there is a statement: 'Such studies have been so far performed only for example cases'. It could be nice with a reference to this. In the ROME metric paper, for example, there is an evaluation of different organization metrics for example cases, so maybe that paper can be referenced here:

https://agupubs.onlinelibrary.wiley.com/doi/full/10.1029/2019JD031801

**Answer**

We thank the referee for the suggestion. The right references have been added to the text.

**Changes in Manuscript (line 26)**

Such studies have been so far performed only for example cases (Retsch et al., 2020; Jin et al., 2022).

Sincerely,

Giulio Mandorli,
Claudia Stubenrauch